# Biparatopic sybodies neutralize SARS-CoV-2 variants of concern and mitigate drug resistance

Justin D Walter[1,†] [iD], Melanie Scherer[2,†] [iD], Cedric A J Hutter[1,†] [iD], Alisa A Garaeva[1,3,†] [iD], Iwan Zimmermann[1,4] [iD], Marianne Wyss[2] [iD], Jan Rheinberger[5] [iD], Yelena Ruedin[6,7] [iD], Jennifer C Earp[1] [iD], Pascal Egloff[1,4] [iD], Michèle Sorgenfrei[1] [iD], Lea M Hürlimann[1] [iD], Imre Gonda[1] [iD], Gianmarco Meier[1] [iD], Sille Remm[1] [iD], Sujani Thavarasah[1] [iD], Geert van Geest[8] [iD], Rémy Bruggmann[8] [iD], Gert Zimmer[6,7] [iD], Dirk J Slotboom[3] [iD], Cristina Paulino[3,5,*] [iD], Philippe Plattet[2,**] [iD] & Markus A Seeger[1,***] [iD]

## Abstract

The ongoing COVID-19 pandemic represents an unprecedented global health crisis. Here, we report the identification of a synthetic nanobody (sybody) pair, Sb#15 and Sb#68, that can bind simultaneously to the SARS-CoV-2 spike RBD and efficiently neutralize pseudotyped and live viruses by interfering with ACE2 interaction. Cryo-EM confirms that Sb#15 and Sb#68 engage two spatially discrete epitopes, influencing rational design of bispecific and tri-bispecific fusion constructs that exhibit up to 100- and 1,000-fold increase in neutralization potency, respectively. Cryo-EM of the sybody-spike complex additionally reveals a novel *up-out* RBD conformation. While resistant viruses emerge rapidly in the presence of single binders, no escape variants are observed in the presence of the bispecific sybody. The multivalent bispecific constructs further increase the neutralization potency against globally circulating SARS-CoV-2 variants of concern. Our study illustrates the power of multivalency and biparatopic nanobody fusions for the potential development of therapeutic strategies that mitigate the emergence of new SARS-CoV-2 escape mutants.

**Keywords** escape mutants; SARS-CoV-2; sybodies; synthetic nanobodies; variants of concern

**Subject Categories** Immunology; Microbiology, Virology & Host Pathogen Interaction; Structural Biology

## Introduction

The spike glycoprotein is the most prominent surface-exposed entity of severe acute respiratory syndrome coronavirus 2 (SARS-CoV-2) and possesses the vital molecular machinery required for recognition and fusion with host membranes (Wrapp *et al*, 2020b). To date, most authorized vaccines against coronavirus disease 2019 (COVID-19) rely on exposure of patients solely to the spike protein (Poland *et al*, 2020). Similarly, the spike protein is the exclusive target of currently approved monoclonal antibody therapies for COVID-19 (Taylor *et al*, 2021). Unfortunately, recent months have seen the emergence and rapid spread of mutant viral strains conferring amino acid changes in the spike protein, which can attenuate neutralization by many convalescent, vaccine-induced, and monoclonal antibodies (Garcia-Beltran *et al*, 2021; Harvey *et al*, 2021). Therefore, from a public health perspective, it is imperative to pursue the development of therapeutic strategies that can withstand the continued emergence of SARS-CoV-2 escape mutants.

The spike protein mutations that cause increased virulence and immune evasion are predominantly found in the receptor-binding domain (RBD) (Piccoli *et al*, 2020; Harvey *et al*, 2021), which is specifically responsible for host recognition via interaction with human angiotensin-converting enzyme 2 (ACE2) (Benton *et al*, 2020). The RBD harbors two hotspots for antibody recognition. One of these epitopes overlaps with the ACE2 binding interface and is evolutionarily unique in SARS-CoV-2; the second so-called "cryptic" epitope is found in a peripheral region that is conserved among RBDs from several characterized coronaviruses (Yuan *et al*, 2020). While individually targeting either epitope with antibodies quickly results in

---

1  Institute of Medical Microbiology, University of Zurich, Zurich, Switzerland
2  Division of Neurological Sciences, Vetsuisse Faculty, University of Bern, Bern, Switzerland
3  Department of Membrane Enzymology at the Groningen Biomolecular Sciences and Biotechnology Institute, University of Groningen, Groningen, The Netherlands
4  Linkster Therapeutics AG, Zurich, Switzerland
5  Department of Structural Biology at the Groningen Biomolecular Sciences and Biotechnology Institute, University of Groningen, Groningen, The Netherlands
6  Institute of Virology and Immunology, Bern & Mittelhäusern, Switzerland
7  Department of Infectious Diseases and Pathobiology, Vetsuisse Faculty, University of Bern, Bern, Switzerland
8  Interfaculty Bioinformatics Unit and Swiss, Institute of Bioinformatics, University of Bern, Bern, Switzerland
    *Corresponding author. Tel: +31 50 363 34 02; E-mail: c.paulino@rug.nl
    **Corresponding author. Tel: +41 31 631 23 70; E-mail: philippe.plattet@vetsuisse.unibe.ch
    ***Corresponding author. Tel: +41 44 634 53 96; E-mail: m.seeger@imm.uzh.ch
    †These authors contributed equally to this work

the emergence of escape mutants (Weisblum *et al*, 2020; Greaney *et al*, 2021), there is growing evidence that simultaneous engagement of both epitopes via polyvalent antibodies may mitigate viral escape (De Gasparo *et al*, 2021; Koenig *et al*, 2021).

Here, we present two synthetic single-domain antibodies (sybodies), designated Sb#15 and Sb#68, that recognize non-overlapping epitopes on the RBD. Sybodies offer several advantages over conventional antibodies such as the potential for rapid development, low-cost production in prokaryotic expression systems, and facile engineering (Zimmermann *et al*, 2018; Jovčevska & Muyldermans, 2020). Cryogenic electron microscopy (cryo-EM) revealed that Sb#15 binds within the ACE2 interface, whereas Sb#68 engages the adjacent conserved cryptic epitope. Structural analysis also demonstrated that the dual presence of Sb#15 and Sb#68 resulted in the adoption of a novel RBD conformation that we termed *up/out*. Fusion of Sb#15 and Sb#68 yielded a bispecific construct, termed GS4, that displayed enhanced avidity and neutralization potency relative to the separate sybodies. Exposure of SARS-CoV-2 to the individual sybodies *in vitro* resulted in the rapid emergence of escape mutants, including a Q493R-RBD variant (within the ACE2 epitope) that has recently been observed in COVID-19 patients treated with a monoclonal antibody (Focosi *et al*, 2021), as well as a novel P384H-RBD mutation in the cryptic epitope. In contrast, no escape mutants were detected upon treatment with GS4. Finally, we found that additional valency engineering via covalent trimerization of GS4, giving a construct we termed Tripod-GS4r, resulted in further enhancement of viral neutralization potential against the B.1.1.7 (Alpha), B.1.351 (Beta) and B.1.617.2 (Delta) SARS-CoV-2 variants of concern (VOCs). Overall, our study demonstrates favorable prospects for such multivalent sybodies to be a valuable therapeutic tool vis-à-vis future SARS-CoV-2 variants or comparable forthcoming viral pandemics.

# Results

### Identification of a sybody pair that (i) simultaneously bind to the spike RBD, (ii) compete with ACE2 interaction and (iii) efficiently neutralize viruses

We sought to engineer a pair of synthetic nanobodies (sybodies), which may mitigate viral escape due to the simultaneous binding of discrete non-overlapping epitopes of the SARS-CoV-2 spike (S) glycoprotein. Using our established sybody generation workflow (Zimmermann *et al*, 2018, 2020), we conducted a selection campaign against the isolated receptor-binding domain (RBD) of the spike protein. Upon screening single sybody clones with ELISA and grating-coupled interferometry (GCI), we identified six sybodies exhibiting affinities against the RBD ranging from 24 to 178 nM (Figs 1A and EV1A, Appendix Table S1). In this study, we focus on sybodies Sb#15 and Sb#68 (Figs 1A and EV1B) that can simultaneously bind the immobilized spike protein (Fig 1B) and exhibit affinities of 12 and 9 nM, respectively, when probing against the entire spike protein stabilized by two prolines (S-2P) (Fig 1A).

To investigate whether Sb#15 and/or Sb#68 could block the interaction between the spike protein and ACE2, we performed an ACE2 competition experiment using GCI. To this end, spike protein was coated on a GCI chip and Sb#15 (200 nM), Sb#68 (200 nM) as

well as a non-randomized convex sybody control (Sb#0, 200 nM) were injected alone or together with ACE2 (100 nM) to monitor binding (Fig 1C). Indeed, Sb#0 did not bind when injected alone and consequently did not disturb ACE2 binding when co-injected. Conversely, both Sb#15 and Sb#68 were found to dominate over ACE2 in the association phase during co-injection, and the resulting curves are highly similar to what was observed when these two sybodies were injected alone. This experiment demonstrated that Sb#15 and Sb#68 compete with ACE2 for access to its binding site on the spike protein.

Having established that Sb#15 and Sb#68 could bind the spike protein and block ACE2 association in vitro, we next asked whether these sybodies (as well as the additional candidate sybodies Sb#16 and Sb#45 Appendix Fig S1, Appendix Table S1) could inhibit the SARS-CoV-2 fusogenic machinery in viral neutralization assays. To varying extents, all assayed sybodies neutralized vesicular stomatitis viruses (VSV) that were pseudotyped with SARS-CoV-2 spike protein (Zettl *et al*, 2020), with estimated $IC_{50}$ values of 2.3 μg/ml (147 nM) and 2.3 μg/ml (138 nM), for Sb#15 and Sb#68, respectively (Fig 2A and B, Table 1). As a positive control for our VSV neutralization assay, we used the previously characterized RBD-binding sybody MR3 (Li *et al*, 2021), which in our assay displayed an $IC_{50}$ value of 0.4 μg/ml, equivalent to the reported value (Li *et al*, 2021). Since Sb#15 and Sb#68 can bind simultaneously to full-length spike protein, we mixed Sb#15 and Sb#68 together to investigate potential additive or synergistic neutralizing activity of these two independent sybodies. Indeed, consistent with the binding assays, the simultaneous presence of both sybodies resulted in slightly improved neutralization profiles with $IC_{50}$ values reaching 1.7 μg/ml (53 nM), suggesting an additive effect. In addition to the individual sybodies, we also explored potential avidity effects of sybodies genetically fused to human IgG1 Fc domains. The respective homodimeric sybody-Fc constructs exhibited VSV pseudotype $IC_{50}$ values of 1.2 μg/ml (16 nM) and 3.9 μg/ml (50 nM) for Sb#15 and Sb#68, respectively (Fig 2C, Table 1). This improvement in VSV neutralization potency suggests that the bivalent arrangement of the Fc fusion constructs resulted in a discernible avidity effect. For neutralization of live SARS-CoV-2 we employed a classical virus neutralization assay and confirmed that both sybodies successfully inhibited cell entry by infectious SARS-CoV-2, with $ND_{50}$ values of 8.8 μg/ml (561 nM) for Sb#15 and 6.3 μg/ml (377 nM) for Sb#68 (Table 1). The approximately 3- to 6-fold discrepancy in neutralization efficacies, measured using either live SARS-CoV-2 virus or pseudotyped VSV, may reflect slight differences in viral physiology (variation of incorporated spikes per viral particle) or could owe to the different assay methods (luciferase emission versus plaque reduction determination). Collectively, these data highlight the successful discovery of a pair of sybodies (Sb#15 and Sb#68) that bind simultaneously to the spike RBD, compete with ACE2 interaction, and neutralize viral infection in vitro.

### Structural basis of Sb#15 and Sb#68 neutralizing activity

To gain structural insights into how Sb#15 and Sb#68 recognize the RBD and neutralize viruses, we performed single-particle cryo-EM analysis of purified sybody-spike protein complexes. Three cryo-EM datasets were collected, allowing a glimpse of the spike protein either simultaneously bound to both sybodies, or associated to

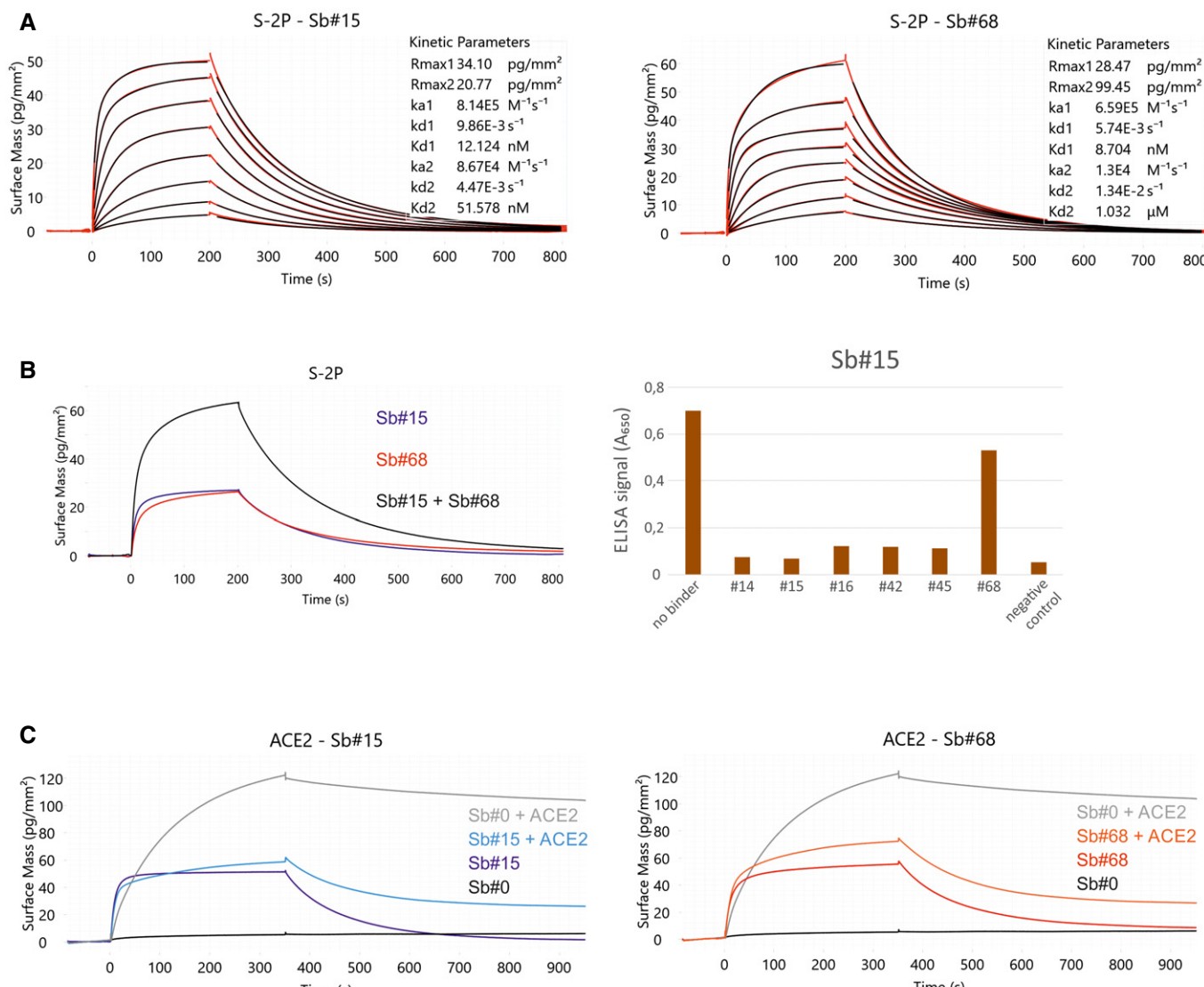

**Figure 1. Sybodies Sb#15 and Sb#68 bind non-overlapping epitopes on the spike protein, and inhibit ACE2 binding.**

A  Affinity determination of Sb#15 and Sb#68 against the immobilized spike protein (S-2P) using GCI. The data were fitted using a heterogeneous ligand model.

B  Left, GCI epitope-binning experiment showing Sb#15 (blue), Sb#68 (red), and their combination (black) against immobilized spike protein (S-2P). Since both sybodies were present at saturating concentrations, the increased amplitude is indicative of simultaneous binding. Right, ELISA experiment confirming dual-binding of Sb#15 and Sb#68. Myc-tagged Sb#15 was immobilized on an anti-myc antibody-coated ELISA plate, followed by exposure of biotinylated RBD, which was premixed with tag-less sybodies (indicated on the x-axis).

C  Competition of sybodies and ACE2 for spike protein binding, investigated by GCI. Biotinylated spike protein was immobilized on the GCI sensor and then Sb#15 (200 nM, left), or Sb#68 (200 nM, right) were injected alone or premixed with human ACE2 (100 nM). Sb#0 represents a non-randomized control sybody.

Sb#15 or Sb#68 alone (Fig 3, Appendix Table S2). The highest resolution was obtained for the spike protein in complex with both sybodies (Figs 3 and EV2, Appendix Fig S1A–K), whereas structures with the individual sybodies were determined based on fewer particles and mainly served to unambiguously assign the binding epitopes of Sb#15 (Appendix Fig S2A–G, Fig EV3A and B) and Sb#68 (Appendix Fig S3A–K, Fig EV4). Analysis of the spike/Sb#15/Sb#68 particles after 3D classification revealed that the spike protein adopts two distinct conformations (Appendix Fig S1F and G). The first conformation (30% of particles) has a three-fold symmetry,

with three RBDs in the *up* conformation (*3up*) and two sybodies bound to each of the RBDs, confirming that Sb#15 and Sb#68 bind simultaneously (Figs 3A and EV2A, Appendix Fig S1F and G).

Although the global resolution of the spike protein in complex with both sybodies is around 3 Å, the local resolution of the RBDs with bound sybodies was only in the range of 6–7 Å, presumably due to conformational flexibility (Appendix Fig S1F and G). Therefore, we did not build a full model for Sb#15, but instead fitted a homology model into the density. In contrast, a crystal structure of Sb#68 in complex with the RBD had recently been determined at a

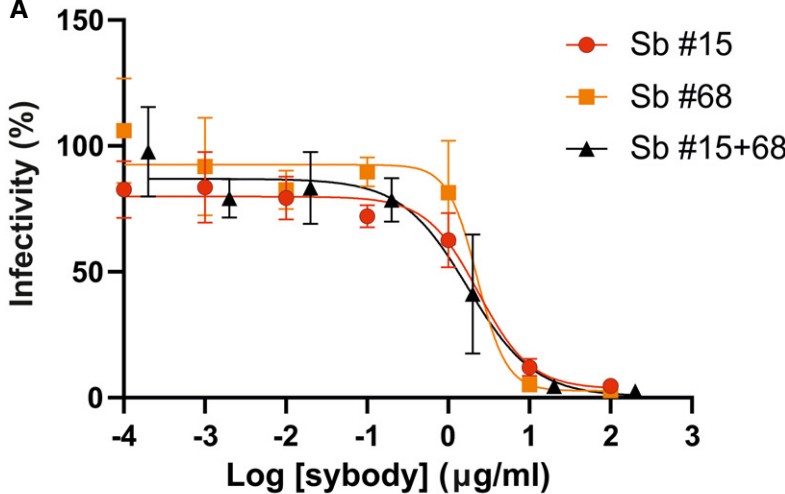

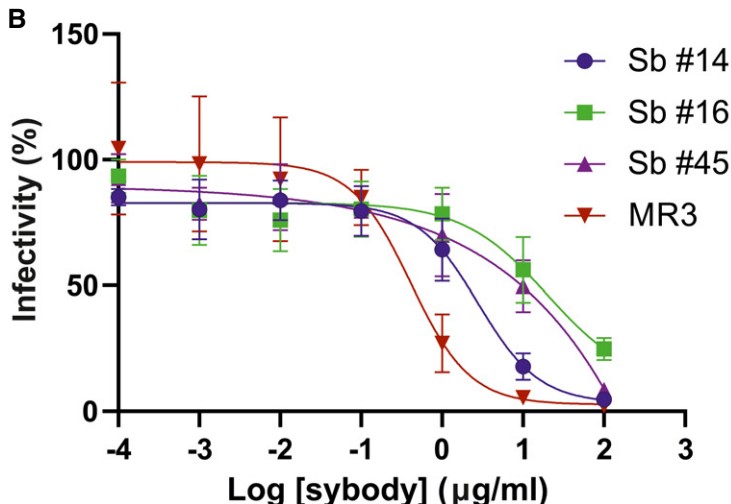

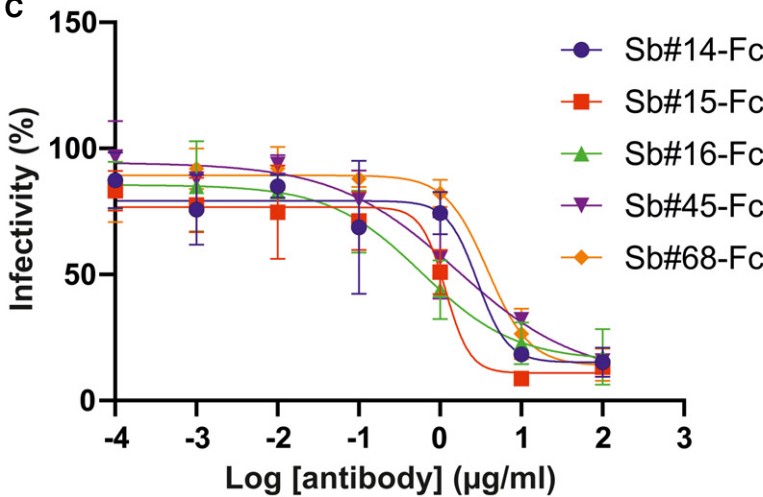

**Figure 2.**

**Figure 2. Neutralization of viral entry using pseudotyped VSVs.**

Neutralization assays using VSVΔG pseudotyped with wild-type SARS-CoV-2 spike protein. Panels show relative infectivity upon exposure to increasing concentrations of the indicated sybody constructs. Error bars correspond to standard deviation of three biological replicates.

A  Sb#15, Sb#68, or an equimolar mixture of both sybodies.
B  Sb#14, Sb#16, Sb#45, or, as a positive control, the previously described sybody MR3.
C  Neutralization by bivalent sybody-Fc fusions.

**Table 1. Summary of neutralization assay results.**

| Binders | SARS-CoV-2 pseudovirus | | Live SARS-CoV-2 | |
| --- | --- | --- | --- | --- |
| | IC$_{50}$ (µg/ml) | IC$_{50}$ (nM) | ND$_{50}$ (µg/ml) | ND$_{50}$ (nM) |
| Sb#14 | 2.8 | 178.3 | nn | nn |
| Sb#15 | 2.3 | 146.5 | 8.8 | 561 |
| Sb#16 | 20 | 1250 | nn | nn |
| Sb#45 | 15 | 910 | nd | nd |
| Sb#68 | 2.3 | 137.7 | 6.3 | 377 |
| Sb#15+Sb#68 | 1.7 | 52.5 | nd | nd |
| GS4 | 0.02 | 0.7 | 0.08 | 2.6 |
| MR3 | 0.4 | 24 | 2.3 | 140 |
| Tripod-GS4r | 0.01 | 0.08 | 0.08 | 0.6 |
| Sb#14-Fc | 2.9 | 37.8 | nd | nd |
| Sb#15-Fc | 1.2 | 15.5 | nd | nd |
| Sb#16-Fc | 0.6 | 7.8 | nd | nd |
| Sb#45-Fc | 1.6 | 20.3 | nd | nd |
| Sb#68-Fc | 3.9 | 49.6 | nd | nd |

nn, non-neutralizing; nd, not determined.

resolution of 2.6 Å (PDB: 7KLW) (Ahmad *et al*, 2021), and therefore, we used this high-resolution structure of Sb#68 to fit it into the cryo-EM density.

Sb#15 binds to the top of the RBD. Its binding epitope roughly consists of two regions (residues 444–448 and 491–507) and thereby strongly overlaps with the ACE2 binding site (Fig 3B). In contrast, Sb#68 binds to the side of the RBD (Appendix Fig S3A–K and Fig EV4) and recognizes a conserved "cryptic" epitope (Yuan *et al*, 2020; Zhou *et al*, 2020) clearly distinct from the ACE2 interaction site, which includes residues 369–385 and R408 and is buried if the RBD is in its *down* conformation. Although the binding epitope of Sb#68 is clearly distinct from the one of ACE2, there would be a steric clash between the Sb#68 backside loops and ACE2, if ACE2 docks to the RBD (Fig 3B). This accounts for Sb#68's ability to compete with ACE2 as evident from GCI analyses (Fig 1C).

The second resolved conformation (20% of particles) of the spike/Sb#15/Sb#68 complex is asymmetric, with the RBDs in three distinct states, and was obtained at a global resolution of 3.3 Å (Figs 3C and EV2B, Appendix Fig S1I). In this conformational state, each RBD was bound to Sb#15, whereas only two RBDs were associated with Sb#68-attributable cryo-EM density. One RBD was in the *up* conformation, having Sb#15 and Sb#68 bound in an analogous fashion as in the symmetric *3up* structure. The orientation of this *up*-RBD closely superimposes with a variety of other reported

cryo-EM structures (Fig 3E). However, interestingly, a second RBD adopted a unique positioning that we term *up-out*. To our knowledge, this conformation has not been observed in prior spike protein structures and therefore represents a novel RBD orientation (Fig 3D and E). Notably, the density for Sb#68 was comparatively weak, indicating either high flexibility or a sub-stoichiometric occupancy. The novel *up-out* conformation appears to be caused by steric influences from the third RBD, which, in a *down* state and singularly bound to Sb#15, acts as a wedge that pushes the second RBD away from the three-fold symmetry axis and into its distinctive orientation (Fig 3C).

Virtually, the same asymmetric *1up/1up-out/1down* spike conformation was observed for the spike/Sb#15 complex, reinforcing our interpretation that wedging by Sb#15 is responsible for the outward movement of the second *up*-RBD (Fig EV3). However, according to our analysis, comprising only a limited number of images (Appendix Fig S3D), Sb#15 alone was unable to induce the *3up* conformation, suggesting that adoption of the *3up* state requires the concerted action of both sybodies to populate this symmetric conformation.

Finally, analysis of the spike/Sb#68 complex dataset revealed two distinct populations (Appendix Figs S4 and S7). The most abundant class showed an *1up/2down* conformation without sybody bound, which is identical to the one obtained for the spike protein alone (Walls *et al*, 2020; Wrapp *et al*, 2020b) (Fig EV4A and B). The second structure featured two RBDs in an *up* conformation with bound Sb#68 (Fig EV4C and D). Density for the third RBD was very weak, presumably due to high intrinsic flexibility, hindering the interpretation of its exact position and conformation. We therefore refer to this conformation as an *2up/1flexible* state. Structural comparisons revealed that Sb#68 cannot access its epitope in the context of the *1up/2down* conformation, due to steric clashes with the neighboring RBD (Fig EV4E). In order to bind, at least two RBDs need to be in the *up* conformation.

## Suppression of emergence of drug-resistant viruses by design of a biparatopic fusion construct

Biochemical and structural data provided evidence that fusing both sybodies may boost viral neutralization. To this end, Sb#15 and Sb#68 were genetically fused via a flexible (GGGGS)$_4$ linker (Fig 4A). The resulting purified bispecific Sb#15-(GGGGS)$_4$-Sb#68 construct, designated as GS4, displayed a ≥ 40-fold increase in binding affinity for the spike protein (apparent $K_d \approx 0.3$ nM), relative to either Sb#15 or Sb#68 alone (Fig 4B). Strikingly, the neutralization potency of GS4 was increased by ≥ 100-fold over the individual binders, for both pseudotyped VSV (IC$_{50}$ = 0.02 µg/ml; 0.7 nM) as well as for live SARS-CoV-2 (ND$_{50}$ = 0.08 µg/ml; 2.6 nM) (Fig 4C, Table 1), which may be attributed to the avidity effect of this biparatopic fusion construct.

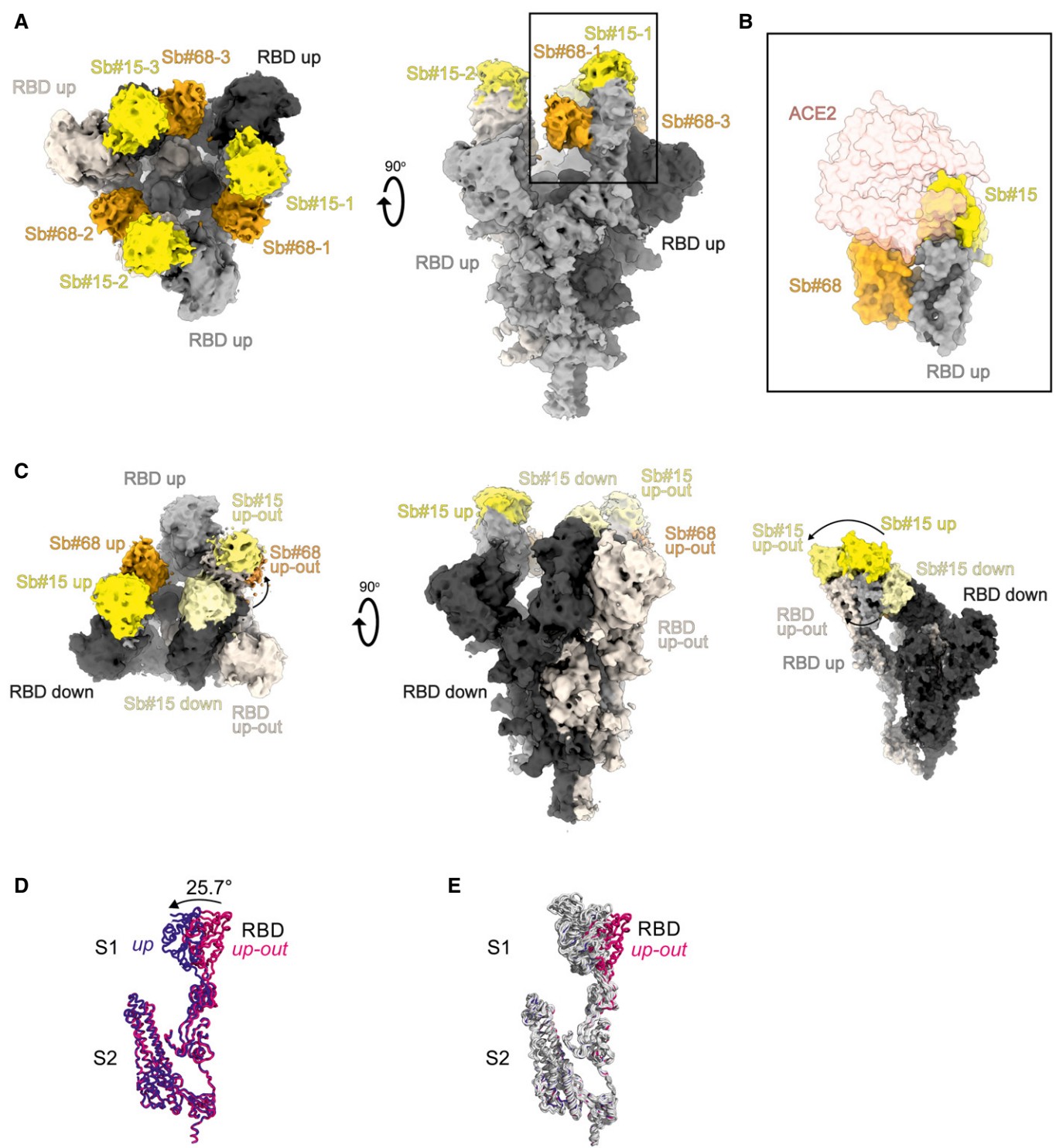

**Figure 3. Cryo-EM maps of S-2P spike in complex with Sb#15 and Sb#68.**

A  Cryo-EM map of S-2P with both Sb#15 and Sb#68 bound to each RBD adopting a symmetrical *3up* conformation.

B  Close-up view showing that ACE2 binding to RBD (PDB ID: 6M0J) is blocked by bound Sb#15 and by a steric clash with Sb#68.

C  Cryo-EM map of S-2P with the three RBDs adopting an asymmetrical *1up/1up-out/1down* conformation. Sb#15 is bound to all three RBDs, while Sb#68 is only bound to the *up* and *up-out* RBD. Final maps blurred to a B factor of −30 Å were used for better clarity of the less resolved RBDs and sybodies. Spike protein is shown in shades of gray, Sb#15 in yellow and Sb#68 in orange.

D  Alignment of structural models for the *up* (blue) and *up-out* (magenta) spike conformations. For clarity, only monomers are shown.

E  The *up-out* RBD conformation is unique among reported spike structures. Superposition of the aligned models (D) with 15 published structures of spike monomers showing *up*-RBDs (gray). PDB identifiers of aligned structures: 6VSB, 6VYB, 6XKL, 6ZGG, 6ZXN, 7A29, 7B18, 7CHH, 7CWT, 7DX9, 7JWB, 7LWW, 7M6F, 7N0H, and 7N1V.

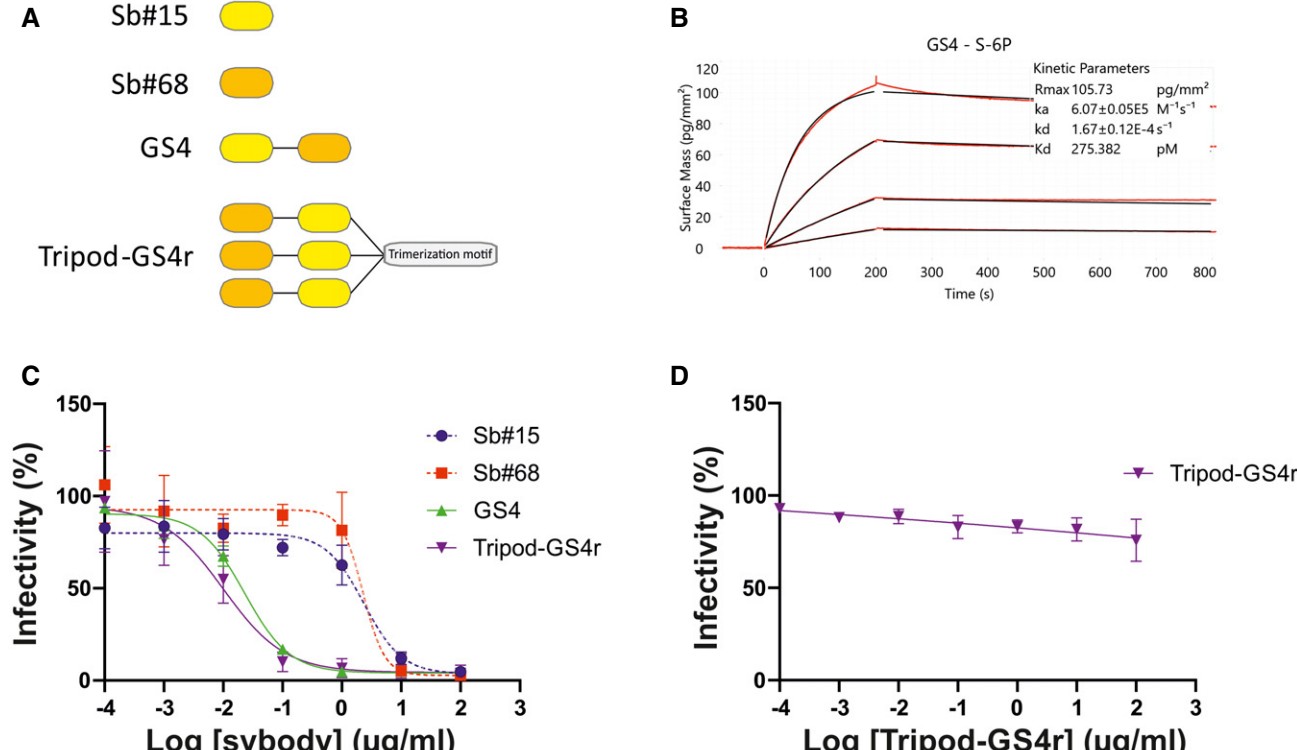

**Figure 4. Fusion of Sb#15 and Sb#68 into biparatopic and multivalent molecules strongly enhance binding affinity and neutralization.**

A  Schematic drawing of biparatopic GS4 and multivalent Tripod-GS4r constructs.
B  Affinity determination of GS4 against immobilized spike protein using GCI.
C  Neutralization assay using VSVΔG pseudotyped with wild-type Spike protein. Relative infectivity was determined in response to increasing GS4 or Tripod-GS4r concentration. The corresponding neutralization data for isolated Sb#15 and Sb#68 are provided as reference.
D  Control experiment using VSVΔG pseudotyped with VSV-G. Infectivity was not affected by Tripod-GS4r. Error bars in panels (C) and (D) correspond to standard deviations of three biological replicates.

We next asked whether simultaneous targeting of two spatially-distinct epitopes within the RBD would provide any advantages toward mitigating the development of escape mutants. By employing a replication-competent VSV*ΔG-S$_{\Delta 21}$ chimera and a reported strategy to generate viral escape (Baum *et al*, 2020), we observed no resistant viruses in GS4-treated cells, whereas escape mutants emerged rapidly in the singular presence of either Sb#15 or Sb#68 (Fig 5A, Table 2). Among the identified mutations from Sb#15- or Sb#68-treated cells, two were selected (Q493R for Sb#15 and P384H for Sb#68) and introduced into isolated RBDs (for binding kinetics measurements) and full-length spike protein (for neutralization determination). Our cryo-EM structures (Fig 3B) as well as a recently determined crystal structure of the RBD/Sb#68 complex (Ahmad *et al*, 2021) are suggestive for a critical impact of both mutations. Sb#15 and Sb#68 exhibited reduced binding with RBD-Q493R and RBD-P384H, respectively, although this attenuation was considerably more pronounced in case of the Sb#15/RBD-Q493R interaction (Fig 5B). This correlated with the severely reduced neutralization efficacy by Sb#15 or Sb#68 against VSV displaying the corresponding adapted spike protein escape variants (Fig 5C, Table 3). A previous study revealed that both mutations were neither significantly affecting ACE2 binding nor the overall expression profile, as revealed by a deep

mutational scanning approach (Starr *et al*, 2020). Finally, the bispecific fusion construct GS4 showed favorable binding kinetics and neutralization profiles in the presence of either individual mutation (Fig 5B and C), supporting the hypothesis that simultaneously targeting multiple epitopes effectively mitigates evolutionary viral adaptation.

## Activity of sybodies and multivalent fusion constructs against globally circulating SARS-CoV-2 variants of concern

We next investigated the efficacy of Sb#15, Sb#68, and the GS4 fusion construct against spike proteins harboring key mutations of the B.1.1.7 (Alpha), B.1.351 (Beta), and B.1.617.2 (Delta) SARS-CoV-2 VOCs. With regard to the RBD, S-Alpha carries the N501Y substitution, spike Beta harbors the combination of K417N, E484K, and N501Y mutations, and spike Delta exhibits the combined L452R and T478K substitutions (Fig 5A).

Consistent with the three spike Beta mutations mapping at, or close to, the RBD/Sb#15 interface (K417N, E484K and N501Y), Sb#15 interacted with all single-mutation variants with reduced affinity, which was most pronounced for RBD-K417N (Fig 6A). Compared to the individual mutations, the combined K417N/E484K/N501Y (KEN) triple mutant displayed a qualitatively additive

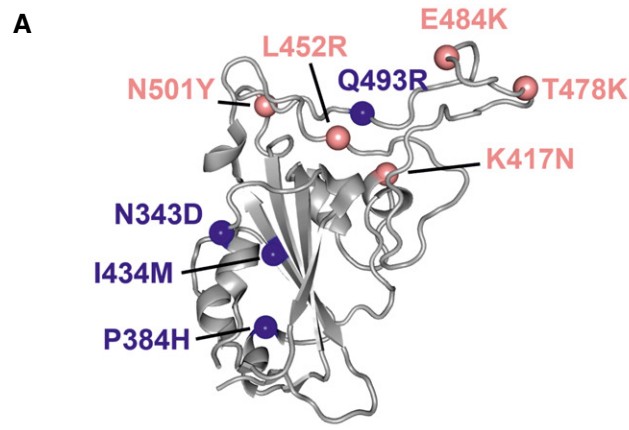

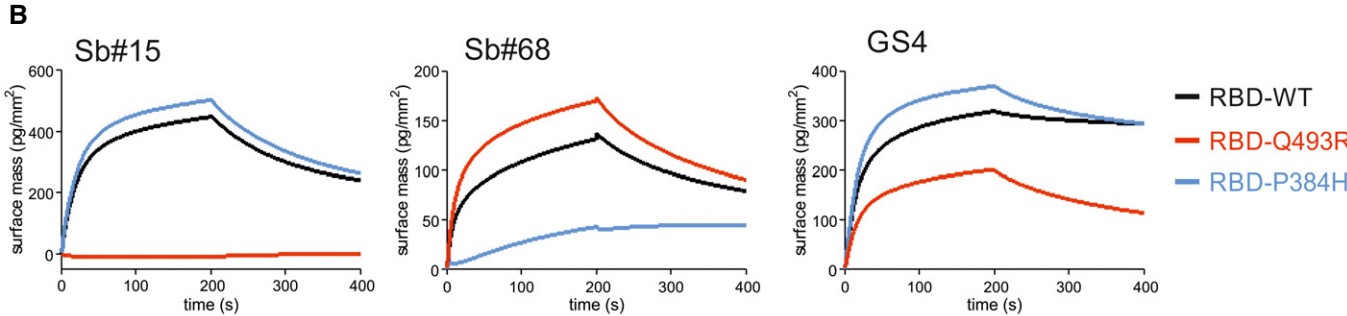

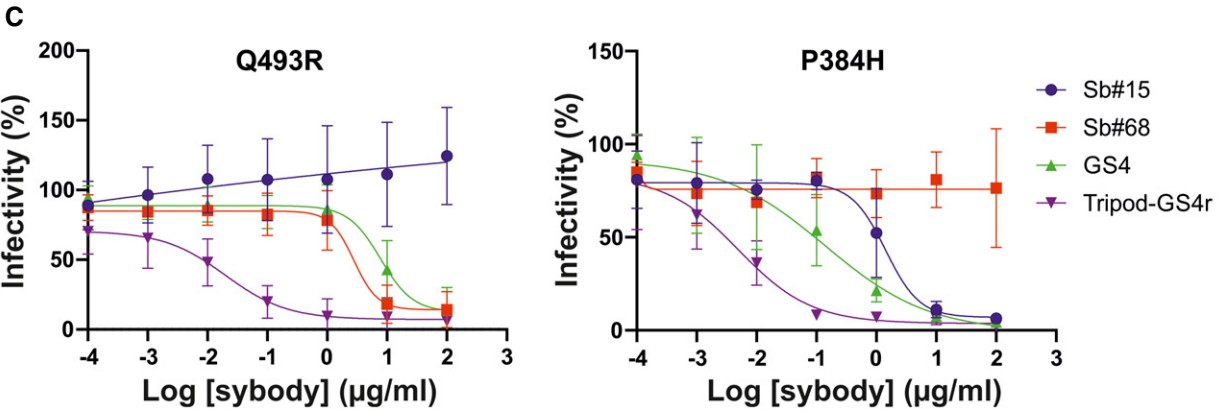

**Figure 5. The bispecific construct GS4 mitigates emergence of novel escape mutants.**

A  Structural context of RBD escape mutants resulting from adaptation experiments in the presence of either Sb#15 or Sb#68 alone (blue spheres). Prominent globally circulating mutations are shown as salmon spheres.

B  GCI-based kinetic analysis of the purified RBD bearing either no mutation (WT), or identified escape mutations Q493R or P384H. Left, middle, and right plots correspond to immobilized Sb#15, Sb#68, or GS4, respectively.

C  Neutralization assay using VSVΔG pseudotyped with Spike protein containing the Q493R or P384H mutation, respectively. Relative infectivity in response to increasing binder concentrations was determined. Error bars correspond to standard deviations of three biological replicates.

effect with regards to reduced binding signal with Sb#15 (Fig 6A). These binding kinetic phenotypes correlated with reduced neutralization efficacies for Sb#15 in pseudotyped VSV experiments with spike Beta mutants (Fig 6B, Table 3). Contrary to the attenuating effects of spike Beta mutations, Sb#15 was not affected by the spike Delta RBD mutations, since both the binding kinetics and neutralizing IC$_{50}$ values remained comparable to WT spike/RBD (Fig 6, Table 3).

**Table 2. Summary of missense escape mutations.**

| Mutation | Nt position (S-gene) | Reference codon | Mutated codon | Spike subdomain | Sb#15 | Sb#68 |
|---|---|---|---|---|---|---|
| A - G | 1027 | aac | gac | S1-RBD | | N343D (91.2[a]) |
| C - A | 1151 | cct | cat | S1-RBD | | P384H (92.7[a]) |
| A - G | 1302 | ata | atg | S1-RBD | | I434 M (93.3[a]) |
| A - G | 1478 | caa | cga | S1-RBD | Q493R (89.2[a]) | |
| C - A | 2903 | tcc | tac | S2 | S968Y (99.1[a]) | |

Nt, nucleotide; RBD, receptor-binding domain.
[a]Percentage of reads carrying the mutation.

**Table 3. Summary of neutralization assay results against specific spike mutants (IC50 in nM).**

| | | | RBD mutants | | | | | | Escape mutants | |
| | | | B.1.351 (Beta) | | | | B.1.617.2 (Delta) | | | |
| | WT | B.1.1.7 (Alpha) | K417N | E484K | N501Y | KEN | L452R/T478K | | Q493R | P384H |
|---|---|---|---|---|---|---|---|---|---|---|
| Sb#15 | 147 | 1,045 | > 5,000 | 841 | 1,026 | > 5,000 | 70 | | > 5,000 | 89 |
| Sb#68 | 138 | 299 | 84 | 156 | 162 | 108 | 90 | | 168 | > 5,000 |
| GS4 | 0.76 | 16 | 7 | 1 | 7 | 436 | 0.66 | | 256 | 5 |
| Tripod-GS4r | 0.08 | 0.09 | 0.04 | 0.11 | 0.12 | 0.08 | 0.16 | | 0.16 | 0.04 |

In sharp contrast to Sb#15, Sb#68, which binds to the peripheral "cryptic epitope" of the RBD that is distant from the investigated mutations, preserved proper binding kinetics and neutralization profiles against all spike variants (Fig 6A, Table 3). Likely owing to this resilience of Sb#68 against the mutations, the binding affinities and neutralizing $IC_{50}$ values of the bispecific GS4 molecule remained very favorable against these individual mutants (Fig 6B, Table 3). However, concerning the combined triple KEN mutant, although no significant impairment in the binding kinetics was recorded (Fig 6A), the neutralizing $IC_{50}$ values revealed that the bispecific construct GS4 was less potent than Sb#68 alone (about 400 nM versus 100 nM, respectively, Table 3). This suggests that, in the context of the KEN mutant, other potential factors that differ between purified spike protein and membrane-anchored spike protein in the context of VSVs influenced the overall neutralization profile of GS4.

We finally determined whether engineering additional layers of multivalency into the bispecific GS4 molecule would provide further advantages in neutralization profiles. To this aim, we grafted trimerization motifs (foldon (McLellan et al, 2013), GCNt (Yin et al, 2006) and a covalently linked trimeric peptide) to the C-terminal region of Sb#15 via a flexible (GGGGS)$_4$ linker (Fig 4A, see Materials and Methods for more details). Furthermore, the N-terminal domain of Sb#15 was fused to the C-terminal region of Sb#68 via a second (GGGGS)$_4$ linker. Of note, the position of both sybodies were reversed as compared to their orientation in GS4; this was based on the observed binding orientation of the sybodies in the context of our cryo-EM structures. This covalently trimerized construct, termed Tripod-GS4r, was confirmed to undergo covalent multimerization that was reversible in reducing conditions (Appendix Fig S4). Neutralization assays were conducted with VSV pseudotyped with wild-type spike or the spike protein harboring key mutations of the Alpha, Beta, and Delta SARS-CoV-2 VOCs (Figs 4C and 6B). Remarkably, the recorded neutralizing $IC_{50}$ values for Tripod-GS4r

were in the low picomolar range against viruses carrying all investigated spike mutations, including the triple KEN (Beta) and double L452R/T478K (Delta) mutants (Table 3). Importantly, confirming specificity and lack of cytotoxicity, Tripod-GS4r did not neutralize control viruses pseudotyped with the non-cognate VSV-G spike glycoprotein (Fig 4D).

## Discussion

Since the start of the SARS-CoV-2 pandemic, multiple viral VOCs have emerged, which threaten the progress made at the vaccination front (Liu et al, 2021a). Therefore, mitigation strategies that take into account the rapid evolution of the virus are urgently needed. Targeting multiple epitopes on the spike protein with large molecules can efficiently mitigate viral escape, a strategy that is already showing promise with recently marketed antibody cocktails (Baum et al, 2020; Copin et al, 2021). Nanobodies offer key advantages over conventional antibodies, in particular, the ease of multimerization, inexpensive production, and high protein stability. The latter simplifies logistics and facilitates development in an inhalable formulation (Van Heeke et al, 2017; Schoof et al, 2020), thereby not only enabling direct delivery to nasal and lung tissues (two key sites of SARS-CoV-2 replication), but also offering the potential of self-administration.

Our study focused on a pair of sybodies, Sb#15 and Sb#68, which recognize two non-overlapping epitopes on the RBD. Both sybodies were found to compete with ACE2 binding. While the binding epitope of Sb#15 directly overlaps with the one of ACE2, this is not the case for Sb#68, which interferes with ACE2 through a steric clash at the sybody backside. Sb#15 and Sb#68 exhibited similar neutralization efficiencies, as well as a moderate synergistic effect in the virus neutralization test when both individual sybodies were mixed

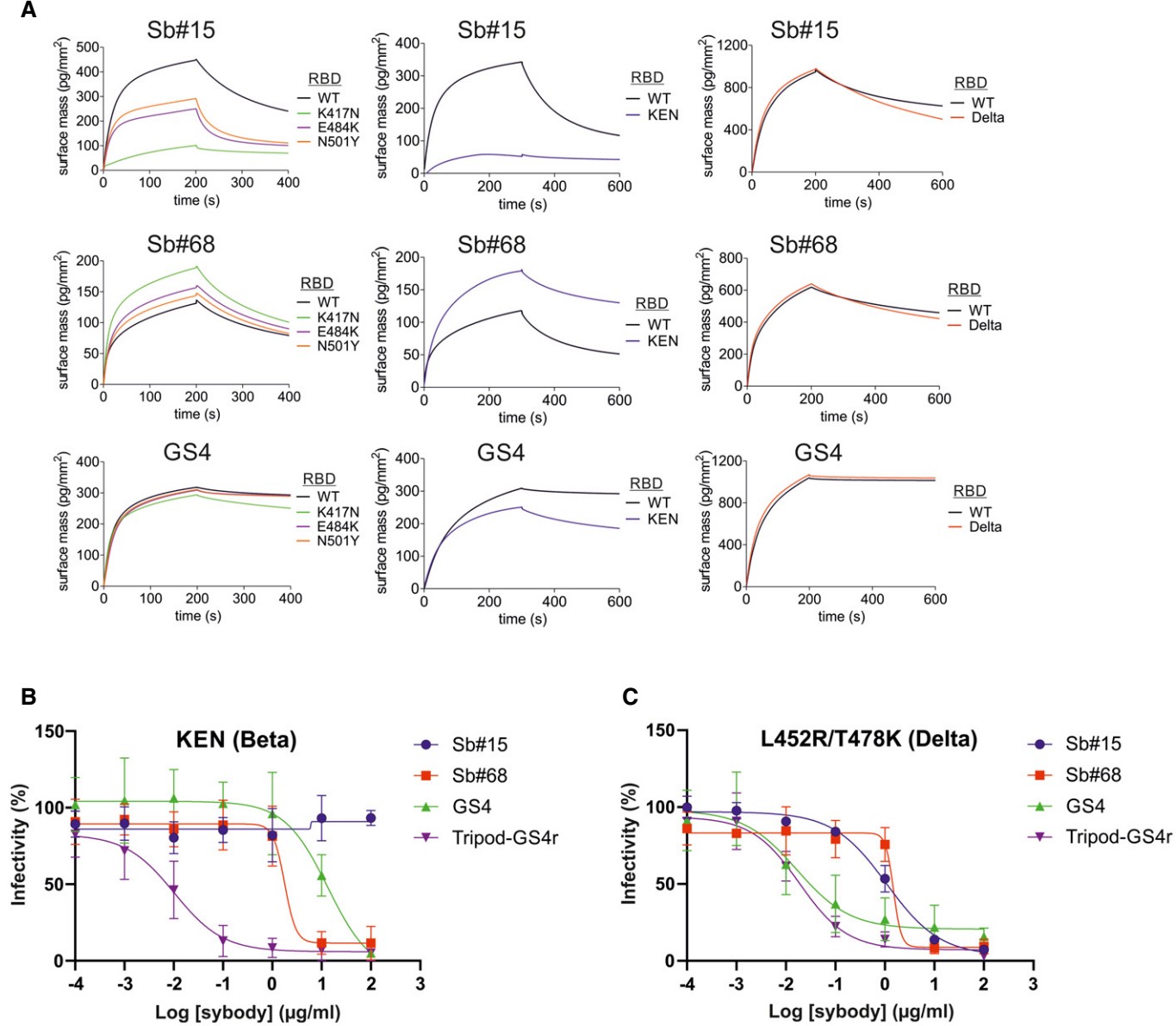

**Figure 6. Affinity and neutralization by sybody constructs for variants of concern.**

A  GCI-based kinetic analysis of interactions between immobilized sybodies (indicated above each plot) and wild-type (WT) or mutant RBDs carrying the individual K417N, E484K, or N501Y mutations (left panels), the combined triple KEN (Beta) mutations (middle panels), or the double L452R/T478K (Delta) mutations (right panels). Sb#15, Sb#68, and GS4 were immobilized on independent flow-cells via biotinylated Avi-tags, and the RBD variants were sequentially injected at a concentration of 200 nM.

B, C  Neutralization assays using VSVΔG pseudotyped with SARS-CoV-2 spike protein containing the triple KEN (Beta) mutations (B) or the double L452R/T478K (Delta) mutations (C). Relative infectivity in response to increasing binder concentrations was determined. Error bars correspond to standard deviations of three biological replicates.

together. This synergy may be explained by the concerted action of the sybodies to compete with ACE2 docking via epitope blockage and steric clashing.

Cryo-EM analyses confirmed distinct binding epitopes for the two sybodies Sb#15 and Sb#68. Without sybodies, the spike protein predominantly assumes an equilibrium between the *3down* and the *1up2down* conformation (Walls *et al*, 2020; Wrapp *et al*, 2020b). Upon addition of Sb#15, the conformational equilibrium was shifted

toward an asymmetric *1up/1up-out/1down* state, whereas addition of Sb#68 favored an asymmetric state with RBDs adopting a *2up/1flexible* conformation. When added together, the sybodies synergized to stabilize two states: a predominant *3up* state and the asymmetric *1up/1up-out/1down* state, thereby shifting the conformational equilibrium of the spike toward RBD conformations competent for ACE2 binding. These structural findings are reminiscent of a recent study, in which a pair of nanobodies isolated from

immune libraries (VHH E and VHH V) was found to bind to similar epitopes as Sb#15 and Sb#68 (Koenig et al, 2021). However, in contrast to Sb#15 stabilizing the asymmetric *1up/1up-out/1down* state when added alone, the corresponding VHH E nanobody is exclusively bound to, and thereby stabilizes, the *3up* conformation. Hence, what is unique for our Sb#15/Sb#68 pair is its concerted action to shift the conformational equilibrium of the spike toward the *3up* state and its capability to trap the spike protein in an unusual *1up/1up-out/1down* conformation, which to the best of our knowledge has not been previously described.

Akin to the antibodies CR3022 and EY6A (Huo et al, 2020b; Zhou et al, 2020) as well as a growing number of nanobodies (Koenig et al, 2021; Pymm et al, 2021; Sun et al, 2021; Xu et al, 2021), the Sb#15/Sb#68 pair stabilized spike conformations with *2up* or *3up* RBDs. Thereby, the spike protein may be destabilized, resulting in the premature and unproductive transitions to the irreversible post-fusion state. This mechanism was dubbed "receptor mimicry" in a study on a neutralizing antibody S230, which only bound to *up*-RBDs and thereby triggered fusogenic conformational changes of SARS-CoV-1 spike (Walls et al, 2019). In elegant experiments, Koenig et al (2021) could demonstrate that stabilization of the spike protein in its *3up* conformation by the addition of nanobodies VHH E and V indeed resulted in aberrant activation of the spike fusion machinery. Hence, it is plausible to assume that our sybody pair inhibits SARS-CoV-2 infection and/or entry via such a receptor mimicry mechanism, in addition to blockage of ACE2 binding.

The binding epitope of Sb#68, also called the "cryptic" epitope (Yuan et al, 2020), is highly conserved between SARS-CoV-1 and SARS-CoV-2. The same conserved epitope is also recognized by the human antibodies CR3022 (isolated from a SARS-CoV-1-infected patient and showing cross-specificity against SARS-CoV-2) and EY6A (Huo et al, 2020b; Zhou et al, 2020) as well as the nanobody VHH-72, which had been originally selected against SARS-CoV-1 but was shown to cross-react with SARS-CoV-2 (Wrapp et al, 2020a) (Fig EV5A–D). Recent months have brought about a growing number of other nanobodies from immune and synthetic libraries whose epitopes overlap with Sb#68 (Koenig et al, 2021; Pymm et al, 2021; Sun et al, 2021; Wagner et al, 2021; Xu et al, 2021; Yao et al, 2021), suggesting that the cryptic epitope constitutes a preferred binding site for VHHs (Fig EV5C). The cryptic epitope remains unchanged in the investigated VOCs, including the B.1.1.7 (Alpha), B.1.351 (Beta), and the B.1.617.2 (Delta) lineages (Fig 5A), and consequently, neutralization efficiency of Sb#68 is unaffected against these variants (Fig 6B and C, Table 3).

Fusion of nanobodies via flexible linkers has emerged as a promising strategy to improve neutralization efficiencies by exploiting avidity effects. This potency-boosting procedure has been explored in the context of SARS-CoV-2 binders by either fusing up to three identical nanobodies (multivalency) (Schoof et al, 2020; Xiang et al, 2020; Xu et al, 2021), or by the structure-based design of biparatopic nanobodies (Wagner et al, 2019; Koenig et al, 2021). We exploited our structural data to first fuse Sb#15 and Sb#68 into the biparatopic GS4 construct, which resulted in a more than 100-fold gain of neutralization efficiency (Table 1). In a subsequent fusion step, the biparatopic GS4 construct was equipped with a trimerization domain (Tripod-GS4r), resulting in a further 10-fold boost in neutralization potential, thereby increasing the cumulative neutralization potency by a factor of around 1,000 when compared to the single constituent nanobodies (Table 1). To our knowledge, engineering of such a "trimer-of-dimers" construct has not yet been attempted with anti-SARS-CoV-2 binder proteins. In addition to neutralizing virus entry via competition with receptor-binding and inducing premature activation of the fusion machinery, the unique multivalent structure of Tripod-GS4r may trigger clustering of neighboring spikes, thereby eventually deactivating multiple viral entry machineries simultaneously.

The ability of enveloped RNA viruses such as SARS-CoV-2 to rapidly develop resistance mutations is a crucial issue of consideration for the development of reliable therapeutics. Escape mutants indeed rapidly emerged when *in vitro* selection experiments were carried with single monoclonal antibodies or nanobodies targeting either the receptor-binding motif or the cryptic epitope alone (Baum et al, 2020; Copin et al, 2021; Koenig et al, 2021). It is interesting to note that the Q493R escape mutation we isolated for Sb#15 has been observed in a COVID-19 patient who received monoclonal antibody therapy (Focosi et al, 2021), and is also present in the recently emergent B.1.1.529 (Omicron) SARS-CoV-2 variant (Liu et al, 2021b). Thus, an attractive strategy to potentially suppress mutational escape is to employ a cocktail of neutralizing antibodies binding to discrete epitopes of the spike. Accumulating evidence indeed demonstrates that rapid viral escape is not observed when experiments are performed either in the presence of a combination of neutralizing monoclonal antibodies/nanobodies or in the presence of biparatopic fusion constructs (Baum et al, 2020; Copin et al, 2021; Koenig et al, 2021). Furthermore, in addition to efficiently suppress mutational escape, our biparatopic molecules (*e.g.*, GS4 and Tripod-GS4r) consistently retained their neutralization capacity against pseudotyped VSV carrying spikes that harbored key mutations of common SARS-CoV-2 VOCs. In particular, the trimerized biparatopic construct (Tripod-GS4r) exhibited ultra-potent neutralizing activity against all tested spike variants, with $IC_{50}$ values of low picomolar range. Hence, our study provides evidence that combining multivalency and biparatopic nanobody fusion proteins represent a promising strategy to potentially generate therapeutic molecules with clinical relevance. While our own Tripod-GS4r construct may cause problems with immune reactivity in a therapeutic setting due to the origin of the utilized multimerization domains, trimerization domains of human origin can be used instead to overcome this potential issue (Guttler et al, 2021).

In conclusion, the rapid selection of sybodies (Zimmermann et al, 2020) and their swift biophysical, structural, and functional characterization provide a foundation for the accelerated reaction to potential future pandemics. In contrast to a number of synthetic or naïve SARS-CoV-2 nanobodies from other libraries that required a post-selection maturation process to reach satisfactory affinities (Huo et al, 2020a; Schoof et al, 2020; Wellner et al, 2021; Zupancic et al, 2021), sybodies selected by us and by other labs (Custodio et al, 2020; Yao et al, 2021) exhibited affinities in the single- and double-digit nM range and where thus of similar affinity as nanobodies isolated from immune libraries using classical phage display (Hanke et al, 2020; Koenig et al, 2021; Wagner et al, 2021). Deep-mining of such sybody pools with our recently described flycode technology is expected to facilitate discovery of exceptional sybodies possessing very slow off-rates or recognizing rare epitopes (Egloff et al, 2019). Single-domain antibodies and their derivative multi-component formats can be produced inexpensively and the

biophysical properties of single-domain antibodies make them feasible for development in an inhalable formulation, thereby offering the potential of self-administration. Hence, nanobodies show great promise to be used as prophylactic agents in the current or future pandemics.

# Materials and Methods

### SARS-CoV-2 expression constructs and commercially acquired proteins

For initial sybody selection experiments and binding affinity measurements, a gene encoding SARS-CoV-2 residues Pro330—Gly526 (RBD, GenBank accession QHD43416.1), downstream from a modified N-terminal human serum albumin secretion signal (Attallah *et al*, 2017), was chemically synthesized (GeneUniversal). This gene was subcloned using FX technology (Geertsma & Dutzler, 2011) into a custom mammalian expression vector (Brunner *et al*, 2014), appending a C-terminal 3C protease cleavage site, myc tag, Venus YFP(Nagai *et al*, 2002), and streptavidin-binding peptide (Keefe *et al*, 2001) onto the open reading frame (RBD-vYFP). A second purified RBD construct, consisting of SARS-CoV-2 residues Arg319—Phe541 fused to a murine IgG1 Fc domain, was purchased from Sino Biological (RBD-Fc, Cat#: 40592-V05H). For kinetic interaction analysis of RBD escape mutants and sybodies, an RBD construct consisting of residues Arg319—Phe541, downstream from the native N-terminal SARS-CoV-2 secretion signal (Met1—Ser13) and appended with a C-terminal 10x-histidine tag (RBD-His), was cloned into a custom mammalian expression vector derived from pcDNA3.1 (Thermo Fisher). Selected mutations P384H, K417N, L452R, T478K, E484K, Q493R, and N501Y were introduced into RBD-His using QuikChange site-directed mutagenesis. Expression plasmids harboring the prefusion ectodomain of the SARS-CoV2 spike protein (Met1—Gln1208), containing two or six stabilizing proline mutations (S-2P or S-6P, respectively) and a C-terminal foldon trimerization motif, HRV 3C protease cleavage site, and twin-strep tag, were a generous gift from Jason McLellan (Hsieh *et al*, 2020; Wrapp *et al*, 2020b). Recombinant human ACE2 was purchased from mybiosource.com (Cat# MBS8248492). Recombinant Fc fusions of sybodies Sb#14, Sb#15, Sb#16, Sb#45, and Sb#68 were produced by Absolute Antibody.

### SARS-CoV-2 protein expression and purification

Suspension-adapted Expi293 cells (Thermo) were transiently transfected using Expifectamine according to the manufacturer protocol (Thermo), and expression was continued for 3–5 days in a humidified environment at 37°C, 8% $CO_2$. Cells were pelleted (500 *g*, 10 min), and culture supernatant was filtered (0.2 μm mesh size) before being incubated with the appropriate affinity chromatography matrix. For RBD-vYFP, NHS-agarose beads covalently coupled to the anti-GFP nanobody 3K1K (Kirchhofer *et al*, 2010) were used for affinity purification, and RBD-vYFP was eluted with 0.1 M glycine, pH 2.5, into tubes that were pre-filled with 1/10 vol 1 M Tris (pH 9.0). Strep-Tactin®XT Superflow® (iba lifesciences) was used to pull down twin-strep-tagged S-2P or S-6P from culture supernatant, followed by elution with 50 mM biotin. Ni-NTA beads were used

for affinity purification of RBD-His, which was eluted with 300 mM imidazole. All affinity-purified SARS-CoV-2 proteins were also subjected to size-exclusion chromatography using either a Superdex 200 Increase 10/300 GL column for RBD constructs, or a Superose 6 Increase 10/300 GL column for spike proteins.

### Sybody selections

Sybody selections, entailing one round of ribosome display followed by two rounds of phage display, were carried out as previously detailed with the three synthetic sybody libraries designated concave, loop and convex (Zimmermann *et al*, 2020). All targets were chemically biotinylated using NHS-Biotin (Thermo Fisher #20217) according to the manufacturer protocol. Binders were selected against two different constructs of the SARS-CoV-2 RBD; an RBD-vYFP fusion and an RBD-Fc fusion. MBP was used as background control to determine the enrichment score by qPCR (Zimmermann *et al*, 2020). In order to avoid enrichment of binders against the fusion proteins (YFP and Fc), we switched the two targets after ribosome display. For the off-rate selections, we did not use non-biotinylated target proteins as described (Zimmermann *et al*, 2020) because we did not have the required amounts of purified target protein. Instead, we employed a pool competition approach. After the first round of phage display, all three libraries of selected sybodies, for both target-swap selection schemes, were subcloned into the pSb_init vector (giving approximately $10^8$ clones) and expressed in *E. coli* MC1061 cells. The resulting three expressed pools were subsequently combined, giving one sybody pool for each selection scheme. These two final pools were purified by Ni-NTA affinity chromatography, followed by buffer exchange of the main peak fractions using a desalting PD10 column in TBS pH 7.5 to remove imidazole. The pools were eluted with 3.2 ml of TBS pH 7.5. These two purified pools were used for the off-rate selection in the second round of phage display at concentrations of approximately 390 μM for selection variant 1 (competing for binding to RBP-Fc) and 450 μM for selection variant 2 (competing for binding to RBP-YFP). The volume used for off-rate selection was 500 μl, with 0.5% BSA and 0.05% Tween-20 added to pools immediately prior to the competition experiment. Off-rate selections were performed for 3 min. For identification of binder hits, ELISAs were performed as described (Zimmermann *et al*, 2020). 47 single clones were analyzed for each library of each selection scheme. Since the RBD-Fc construct was incompatible with our ELISA format due to the inclusion of Protein A to capture an α-myc antibody, ELISA was performed only for the RBD-vYFP (50 nM) and the M) and later on with the S-2P (25 nM). Of note, the three targets were analyzed in three separate ELISAs. As negative control to assess background binding of sybodies, we used biotinylated MBP (50 nM). 72 positive ELISA hits were sequenced (Microsynth, Switzerland).

### Expression and purification of sybodies

The 63 unique sybodies were expressed and purified as described (Zimmermann *et al*, 2020). In short, all 63 sybodies were expressed overnight in *E. coli* MC1061 cells in 50-ml cultures. The next day, the sybodies were extracted from the periplasm and purified by Ni-NTA affinity chromatography (batch binding) followed by size-exclusion chromatography using a Sepax SRT-10C SEC100

size-exclusion chromatography (SEC) column equilibrated in TBS, pH 7.5, containing 0.05% (v/v) Tween-20 (detergent was added for subsequent kinetic measurements). Six out of the 63 binders (Sb#4, Sb#7, Sb#18, Sb#34, Sb#47, and Sb#61) were excluded from further analysis due to suboptimal behavior during SEC analysis (i.e., aggregation or excessive column matrix interaction).

## Generation of bispecific sybody fusions

To generate the bispecific sybodies (Sb#15-Sb#68 fusion with variable glycine/serine linkers), Sb#15 was amplified from pSb-init_Sb#15 (Addgene #153523) using the forward primer 5′-ATA TAT GCT CTT CAA GTC AGG TTC and the reverse primer 5′-TAT ATA GCT CTT CAA GAA CCG CCA CCG CCG CTA CCG CCA CCA CCT GCG CTC ACA GTC AC, encoding 2× a GGGGS motif, followed by a SapI cloning site. Sb#68 was amplified from pSb-init_Sb#68 (Addgene #153527) using forward primer 5′-ATA TAT GCT CTT CTT CTG GTG GTG GCG GTA GCG GCG GTG GCG GTA GTC AAG TCC AGC TGG TGG combined with the reverse primer 5′-TAT ATA GCT CTT CCT GCA GAA AC. The forward primers start with a SapI site (compatible overhang to Sb#15 reverse primer), followed by 2× the GGGGS motif. The PCR product of Sb#15 was cloned in frame with each of the three PCR products of Sb#68 into pSb-init using FX-cloning (Geertsma & Dutzler, 2011), thereby resulting in three fusion constructs with linkers containing 4× GGGGS motives as flexible linker between the sybodies (called GS4). The bispecific fusion construct GS4 was expressed and purified the same way as single sybodies (Zimmermann *et al*, 2020).

## Construction, expression, and purification of Tripod-GS4r

In order to engineer a trivalent GS4 molecule, we fused the following elements (from N- to C-terminus): Sb#68-(GGGGS)$_4$-Sb#15-(GGGGS)$_4$-CC-GCNt-Foldon-TST. CC is a peptide derived from the CDV F protein (589-599) and contains two successive cysteine mutations (I595C and L596C) shown in the context of soluble measles virus F protein to stabilize the trimeric prefusion state (Hashiguchi *et al*, 2018). GCNt is a well-known trimerization motif that was previously described (Yin *et al*, 2006). Foldon stems from fibritin (McLellan *et al*, 2013). TST denotes a C-terminal His/TwinStrepTag sequence for purification purposes. The Tripod-GS4r expression plasmid (3 mg) was sent to the Protein Production and Structure Core Facility of the EPFL (Switzerland) for expression (7 days in ExpiCHO cells). Subsequently, the protein was purified from 1 l of supernatant using a 5 ml StrepTtrapXT column (Cytavia) and eluted with 500 mM biotin (Cytivia).

## Dual-sybody competition ELISA

Purified sybodies carrying a C-terminal myc-His Tag (Sb_init expression vector) were diluted to 25 nM in 100 µl PBS pH 7.4 and directly coated on Nunc MaxiSorp 96-well plates (Thermo Fisher #44-2404-21) at 4°C overnight. The plates were washed once with 250 µl TBS pH 7.5 per well followed by blocking with 250 µl TBS pH 7.5 containing 0.5% (w/v) BSA per well. In parallel, chemically biotinylated prefusion Spike protein (S-2P) at a concentration of 10 nM was incubated with 500 nM sybodies for 1 h at room temperature in TBS-BSA-T. The plates were washed three times with 250 µl TBS-T

per well. Then, 100 µl of the S-2P-sybody mixtures were added to the corresponding wells and incubated for 3 min, followed by washing three times with 250 µl TBS-T per well. 100 µl Streptavidin-peroxidase polymer (Merck, Cat#S2438) diluted 1:5,000 in TBS-BSA-T was added to each well and incubated for 10 min, followed by washing three times with 250 µl TBS-T per well. Finally, to detect S-2P bound to the immobilized sybodies, 100 µl ELISA developing buffer (prepared as described previously (Zimmermann *et al*, 2020)) was added to each well, incubated for 1 h (due to low signal) and absorbance was measured at 650 nm. As a negative control, TBS-BSA-T devoid of protein was added to the corresponding wells instead of a S-2P-sybody mixture.

## Grating-coupled interferometry (GCI)

Kinetic characterization of sybodies binding onto SARS-CoV-2 spike proteins was performed using GCI on the WAVEsystem (Creoptix AG, Switzerland), a label-free biosensor. For the off-rate screening, biotinylated RBD-vYFP and ECD were captured onto a Streptavidin PCP-STA WAVEchip (polycarboxylate quasi-planar surface; Creoptix AG) to a density of 1,300–1,800 pg/mm$^2$. Sybodies were first analyzed by an off-rate screen performed at a concentration of 200 nM to identify binders with sufficiently high affinities. The six sybodies Sb#14, Sb#15, Sb#16, Sb#42, Sb#45, and Sb#68 were then injected at increasing concentrations ranging from 1.37 nM to 1 µM (three-fold serial dilution, 7 concentrations) in 20 mM Tris pH7.5, 150 mM NaCl supplemented with 0.05% Tween-20 (TBS-T buffer). Sybodies were injected for 120 s at a flow rate of 30 µl/min per channel, and dissociation was set to 600 s to allow the return to baseline.

In order to determine the binding kinetics of Sb#15 and Sb#68 against intact spike proteins, the ligands RBD-vYFP, S-2P and S-6P were captured onto a PCP-STA WAVEchip (Creoptix AG) to a density of 750 pg/mm$^2$, 1,100 pg/mm$^2$, and 850 pg/mm$^2$, respectively. Sb#15 and Sb#68 were injected at concentrations ranging from 1.95 to 250 nM or 3.9 to 500 nM, respectively (2-fold serial dilution, 8 concentrations) in TBS-T buffer. Sybodies were injected for 200 s at a flow rate of 80 µl/min and dissociation was set to 600 s. In order to investigate whether Sb#15 and Sb#68 bind simultaneously to the RBD, S-2P, and S-6P, both binders were either injected alone at a concentration of 200 nM or mixed together at the same individual concentrations at a flow rate of 80 µl/min for 200 s in TBS-T buffer.

To measure binding kinetics of the three bispecific fusion construct GS4, S-6P was captured as described above to a density of 1,860 pg/mm$^2$ and increasing concentrations of the bispecific fusion constructs ranging from 1 to 27 nM (3-fold serial dilution and 4 concentrations) in TBS-T buffer at a flow rate of 80 µl/min. Because of the slow off-rates, we performed a regeneration protocol by injecting 10 mM glycine pH 2 for 30 s after every binder injection.

For ACE2 competition experiments, S-2P was captured as described above. Then, Sb#15, Sb#68, and Sb#0 (non-randomized convex sybody control) were either injected individually or premixed with ACE2 in TBS-T buffer. Sybody concentrations were at 200 nM and ACE2 concentration was at 100 nM.

For semi-quantitative kinetic analysis of interactions between sybodies and RBD mutants, C-terminally avi-tagged variants of Sb#15, Sb#68, and GS4 were enzymatically biotinylated (Fairhead & Howarth, 2015) and captured on separate channels of a PCP-STA

WAVEchip (Creoptix AG) to densities of approximately 1,000–3,000 pg/mm$^2$. Purified RBD mutants were diluted to 200 nM in TBS-T buffer, sequentially injected to the GCI system, and association/dissociation signals were recorded.

All sensorgrams were recorded at 25°C and the data analyzed on the WAVEcontrol (Creoptix AG). Data were double-referenced by subtracting the signals from blank injections and from the reference channel. A Langmuir 1:1 model was used for data fitting with the exception of the Sb#15 and Sb#68 binding kinetics for the S-2P and the S-6P spike, which were fitted with a heterogeneous ligand model as mentioned in the main text. For analysis of interactions between sybodies and RBD mutants, no models provided sufficient fits allowing extraction of kinetic parameters, so curves were interpreted via qualitative comparison with wild-type RBD signals.

## SARS-CoV-2 pseudovirus neutralization

Pseudovirus neutralization assays have been previously described (Pallesen *et al*, 2017; Wrapp *et al*, 2020a; Zettl *et al*, 2020). Briefly, propagation-defective, spike protein-pseudotyped vesicular stomatitis virus (VSV) was produced by transfecting HEK-239T cells with SARS-CoV-2 Sdel 18 (SARS-2 S carrying an 18 aa cytoplasmic tail truncation) as described previously (Wang *et al*, 2020). The cells were further inoculated with glycoprotein G *trans*-complemented VSV vector (VSV*ΔG(Luc)) encoding enhanced green fluorescence protein (eGFP) and firefly luciferase reporter genes but lacking the glycoprotein G gene (Berger Rentsch & Zimmer, 2011). After 1-h incubation at 37°C, the inoculum was removed and the cells were washed once with medium and subsequently incubated for 24 h in medium containing 1:3,000 of an anti-VSV-G mAb I1 (ATCC, CRL-2700$^{TM}$). Pseudotyped particles were then harvested and cleared by centrifugation.

For the SARS-CoV-2 pseudotype neutralization experiments, pseudovirus was incubated for 30 min at 37°C with different dilutions of purified sybodies, sybdody fusions, or sybody-Fc fusions. Subsequently, S protein-pseudotyped VSV*ΔG(Luc) was added to Vero E6 cells grown in 96-well plates (25,000 cells/well). At 24 h post-infection, luminescence (firefly luciferase activity) was measured using the ONE-Glo Luciferase Assay System (Promega) and Cytation 5 cell imaging multi-mode reader (BioTek).

## SARS-CoV-2 neutralization test

The serial dilutions of control sera and samples were prepared in quadruplicates in 96-well cell culture plates using DMEM cell culture medium (50 µl/well). To each well, 50 µl of DMEM containing 100 tissue culture infectious dose 50% (TCID$_{50}$) of SARS-CoV-2 (SARS-CoV-2/München-1.1/2020/929) was added and incubated for 60 min at 37°C. Subsequently, 100 µl of Vero E6 cell suspension (100,000 cells/ml in DMEM with 10% FBS) was added to each well and incubated for 72 h at 37°C. The cells were fixed for 1 h at room temperature with 4% buffered formalin solution containing 1% crystal violet (Merck, Darmstadt, Germany). Finally, the microtiter plates were rinsed with deionized water and immune serum-mediated protection from cytopathic effect was visually assessed. Neutralization doses 50% (ND$_{50}$) values were calculated according to the Spearman and Kärber method.

## Generation of chimeric VSV*ΔG-S$_{Δ21}$

Recently, we generated a recombinant chimeric VSV, VSV*ΔG (MERS-S), in which the VSV glycoprotein (G) gene was replaced by the full-length MERS-CoV spike protein (Pfaender *et al*, 2020). VSV*ΔG(MERS-S) also encoded a GFP reporter which was expressed from an additional transcription unit located between the MERS-CoV spike and VSV L genes. In order to generate a chimeric VSV expressing the SARS-CoV-2 spike protein, the MERS-S gene in the antigenomic plasmid pVSV*ΔG(MERS-S) was replaced by a modified SARS-CoV-2 spike gene (Genscript, Piscataway, USA) taking advantage of the flanking *Mlu*I and *Bst*EII endonuclease restriction sites. The modified SARS-CoV-2 spike gene was based on the Wuhan-Hu-1 strain (GenBank accession number NC_045512) but lacked the region encoding the C-terminal 21 amino acids in order to enhance cell surface transport of the spike protein and its incorporation into the VSV envelope (Dieterle *et al*, 2020). In addition, the modified spike gene contained the mutations R685G, H655Y, D253N, W64R, G261R, and A372T, which have been previously reported to accumulate during passaging chimeric VSV-SARS-CoV-2-S on Vero E6 cells (Dieterle *et al*, 2020). The amino acid substitution R685G is located in the S1/S2 proteolytic cleavage site and has been shown to reduce syncytia formation and to enhance virus titers (Dieterle *et al*, 2020).

The chimeric virus was rescued following transfection of cDNA according to a previously described protocol (Kalhoro *et al*, 2009). Briefly, BHK-21 cells were infected with a modified virus Ankara (MVA) expressing T7 bacteriophage RNA polymerase (Sutter *et al*, 1995) using a multiplicity of infection of 1 focus-forming unit (ffu)/cell. Subsequently, the cells were transfected with the VSV antigenomic plasmid along with plasmids driving the T7 RNA polymerase-mediated transcription of the VSV N, P, and L genes. After 24 h of incubation, the cells were trypsinized and seeded along with an equal number of Vero E6 cells and incubated for an additional 48 h at 37°C. The expression of the GFP reporter in the cells was monitored by fluorescence microscopy. The recombinant virus was rescued from the supernatant of GFP-positive cells and passaged subsequently on Vero E6 cells. Following the fourth passage, virus was harvested from the cell culture supernatant and stored in aliquots at −70°C in the presence of 10% fetal bovine serum (FBS).

Infectious virus titers were determined on confluent Vero E6 cells grown in 96-well microtiter plates. The cells were inoculated in duplicate with 40 µl per well of serial 10-fold virus dilutions for 1 h at 37°C. Thereafter, 160 µl of EMEM containing 1% methylcellulose was added to each well, and the cells were incubated for 24 h at 37°C. The number of infectious foci was determined under the fluorescence microscope taking advantage of the GFP reporter and infectious virus titers were calculated and expressed as ffu/ml.

## Selection of VSV*ΔG-S$_{Δ21}$ escape mutants

The selection of VSV*ΔG-S$_{Δ21}$ mutants, which escaped sybody-mediated inhibition, was performed according to a recently described procedure (Baum *et al*, 2020). Briefly, a total of 10$^4$ ffu of the parental VSV*ΔG-S$_{Δ21}$ were incubated with serially diluted sybodies prior to infection of Vero E6 cells that were grown in 24-well cell culture plates. Two days post-infection, the cell culture supernatant from wells containing the highest antibody concentration,

which did not completely inhibit virus replication as monitored by GFP fluorescence was collected and subjected to a second round of selection on Vero E6 cells grown in 96-well microtiter plates in the presence of increasing sybody concentrations. Virus recovered after a third round of selection was used to infect Vero E6 cells grown in 6-well plates.

## NGS analysis of escape mutants

Vero E6 cells were lysed 24 h post-infection with TRIZol reagent (Ambion, Life Technologies, Zug, Switzerland) and total RNA was subsequently isolated using a Direct-zol™ RNA MicroPrep Kit (Zymo Research #R2060) according to the manufacturer's protocol. The recommended DNase treatment was included. The quantity and quality of the extracted RNA was assessed using a Thermo Fisher Scientific Qubit 4.0 fluorometer with the Qubit RNA BR Assay Kit (Thermo Fisher Scientific, Q10211) and an Advanced Analytical Fragment Analyzer System using a Fragment Analyzer RNA Kit (Agilent, DNF-471), respectively. Prior to cDNA library generation, probe-based depletion of ribosomal RNA was performed on 500 ng of total RNA using a RiboCop rRNA Depletion Kit-Human/Mouse/Rat plus Globin (Lexogen #145.96) according to the producer's protocol. Thereafter, the remaining RNA was used as input for a CORALL Total RNA-Seq Library Prep Kit (Lexogen #117.96) in combination with Lexogen workflow A unique dual indexes Set A1 (Lexogen UDI12A_0001-0096) following the corresponding user guide (Lexogen document 117UG228V0200). The quantity and quality of the generated NGS libraries were evaluated using a Thermo Fisher Scientific Qubit 4.0 fluorometer with the Qubit dsDNA HS Assay Kit (Thermo Fisher Scientific, Q32854) and an Advanced Analytical Fragment Analyzer System using a Fragment Analyzer NGS Fragment Kit (Agilent, DNF-473), respectively. Pooled cDNA libraries were paired-end sequenced using a NovaSeq 6000 SP reagent kit v1.5, 300 cycles (illumina 20028402) on an Illumina NovaSeq 6000 instrument. The quality of the sequencing runs was assessed using illumina Sequencing Analysis Viewer (illumina version 2.4.7), and all base call files were demultiplexed and converted into FASTQ files using illumina bcl2fastq conversion software v2.20. The RNA quality-control assessments, generation of libraries, and sequencing runs were performed at the Next Generation Sequencing Platform, University of Bern, Switzerland.

## Bioinformatic analysis

Reads were trimmed using trimmomatic (Bolger *et al*, 2014) version 0.36, while setting the ILLUMINACLIP parameter to 2:30:10, SLIDINGWINDOW to 4:5 and MINLEN to 50. The parameter HEADCROP was set to 12 to remove the UMI sequence at the forward reads and low-quality bases due to random priming of the reverse read. Trimmed reads were aligned against the spike construct sequence (encoding for the mutations R685G, H655Y, D253N, W64R, G261R, and A372T) by using minimap2 (Li, 2018) with the short read mode (-x sr). Read duplicates were marked in the alignment file with samtools markdup (Li *et al*, 2009). Variants were called on each sample separately using gatk4 HaplotypCaller (Poplin *et al*, 2018) with setting the option --sample-ploidy to 1 and --emit-ref-confidence to GVCF. The resulting GVCF files were combined into a single VCF file with gatk GenomicsDBImport and GenotypeGVCFs. The alleles and

counts per allele were reported for each variant for each sample by using bcftools (Li *et al*, 2009) query.

## Cryo-EM sample preparation and data collection

Freshly purified S-2P was incubated with a 1.3-fold molar excess per spike monomer of Sb#15 alone or with Sb#15 and Sb#68 and subjected to size-exclusion chromatography to remove excess sybody. In analogous way, the sample of S-6P with Sb#68 was prepared. The protein complexes were concentrated to 0.7–1 mg/ml using an Amicon Ultra-0.5 ml concentrating device (Merck) with a 100 kDa filter cutoff. 2.8 μl of the sample was applied onto the holey-carbon cryo-EM grids (Au R1.2/1.3, 300 mesh, Quantifoil), which were prior glow discharged at 5–15 mA for 30 s, blotted for 1–2 s and plunge frozen into a liquid ethane/propane mixture with a Vitrobot Mark IV (Thermo Fisher) at 15°C and 100% humidity. Samples were stored in liquid nitrogen until further use. Screening of the grid for areas with best ice properties was done with the help of a home-written script to calculate the ice thickness (manuscript in preparation). Cryo-EM data in selected grid regions were collected in-house on a 200-keV Talos Arctica microscope (Thermo Fisher Scientifics) with a post-column energy filter (Gatan) in zero-loss mode, with a 20-eV slit and a 100 μm objective aperture. Images were acquired in an automatic manner with SerialEM on a K2 summit detector (Gatan) in counting mode at ×49,407 magnification (1.012 Å pixel size) and a defocus range from −0.9 to −1.9 μm. During an exposure time of 9 s, 60 frames were recorded with a total exposure of about 53 electrons/$\text{Å}^2$. On-the-fly data quality was monitored using FOCUS (Biyani *et al*, 2017).

## Image processing

For the S-2P/Sb#15/ Sb#68 complex dataset, in total 14,883 micrographs were recorded. Beam-induced motion was corrected with MotionCor2_1.2.1 (Zheng *et al*, 2017) and the CTF parameters estimated with ctffind4.1.13 (Rohou & Grigorieff, 2015). Recorded micrographs were manually checked in FOCUS (1.1.0), and micrographs, which were out of defocus range (< 0.4 and >2 μm), contaminated with ice or aggregates, and with a low-resolution estimation of the CTF fit (> 5 Å), were discarded. 637,105 particles were picked from the remaining 12,454 micrographs by crYOLO 1.7.5 (Wagner *et al*, 2019), and imported in cryoSPARC v2.15.0 (Punjani *et al*, 2017) for 2D classification with a box size of 300 pixels. After 2D classification, 264,082 particles were imported into RELION-3.0.8 (Zivanov *et al*, 2018) and subjected to a 3D classification without imposed symmetry, where an ab-initio generated map from cryoSPARC low-pass filtered to 50 Å was used as reference. Two classes resembling spike protein revealed two distinct conformations. One class shows a symmetrical state with all RBDs in an *up* conformation (*3up*) and both sybodies bound to each RBD (78,933 particles, 30%). In the asymmetrical class (52,839 particles, 20%) the RBDs adopt one *up*, one *up-out* and one *down* conformation (*1up/1up-out/1down*), where both sybodies are bound to RBDs *up* and *up-out* state, while only Sb#15 is bound to the *down* RBD. The *3up* class was further refined with C3 symmetry imposed. The final refinement, where a mask was included in the last iteration, provided a map at 7.6 Å resolution. Six rounds of per-particle CTF refinement with beam tilt estimation and re-extraction of particles with a box size of 400 pixels improved

resolution further to 3.2 Å. The particles were then imported into cryoSPARC, where non-uniform refinement improved the resolution to 3 Å. The asymmetrical *1up/1up-out/1down* was refined in an analogous manner with no symmetry imposed, resulting in a map at 7.8 Å resolution. Six rounds of per-particle CTF refinement with beam tilt estimation improved resolution to 3.7 Å. A final round of non-uniform refinement in cryoSPARC yielded a map at 3.3 Å resolution. Local resolution estimations were determined in cryoSPARC. All resolutions were estimated using the 0.143 cutoff criterion (Rosenthal & Henderson, 2003) with gold-standard Fourier shell correlation (FSC) between two independently refined half-maps (Scheres & Chen, 2012). The directional resolution anisotropy of density maps was quantitatively evaluated using the 3DFSC web interface (https://3dfsc.salk.edu) (Tan *et al*, 2017).

A similar approach was performed for the image processing of the S-2P/Sb#15 complex. In short, 2,235 micrographs were recorded, and 1,582 used for image processing after selection. 66,632 particles were autopicked via crYOLO and subjected to 2D classification in cryoSPARC. 57,798 selected particles were used for subsequent 3D classification in RELION-3.0.8, where the symmetrical *3up* map, described above, was used as initial reference. The best class comprising 22,055 particles (38%) represented an asymmetrical *1up/1up-out/1down* conformation with Sb#15 bound to each RBD. Several rounds of refinement and CTF refinement yielded a map of 4.0 Å resolution.

For the dataset of the S-6P/Sb#68 complex, in total 5,109 images were recorded, with 4,759 used for further image processing. 344,976 particles were autopicked via crYOLO and subjected to 2D classification in cryoSPARC. 192,942 selected particles were imported into RELION-3.0.8 and used for subsequent 3D classification, where the symmetrical *3up* map, described above, was used as initial reference. Two distinct classes of spike protein were found. One class (24,325 particles, 13%) revealed a state in which two RBDs adopt an *up* conformation with Sb#68 bound, whereby the density for the third RBD was poorly resolved representing an undefined state. Several rounds of refinement and CTF refinement yielded a map of 4.8 Å resolution. Two other classes, comprising 44,165 particles (23%) and 84,917 particles (44%), were identical. They show a *1up/2down* configuration without Sb#68 bound to any of the RBDs. Both classes were processed separately, whereby the class with over 80k particles yielded the best resolution of 3.3 Å and was used for further interpretation. A final non-uniform refinement in cryoSPARC further improved resolution down to 3.1 Å.

### Model building

Model building was carried out in COOT [72] using previously determined SARS-CoV-2 spike protein structures (PDB ID 7MY2, 6ZGG), Sb#68 crystal structure (PDB ID 7KLW) as reference and anti-GFP nanobody structure (PDB ID 3K1K) as a homology model for Sb#15. After each round of real-space refinement performed in Phenix [73], coordinates were manually inspected and edited in COOT and submitted to another refinement round in an iterative way.

### Biosafety

Work with infectious SARS-CoV-2 has been approved by the Swiss Federal Office of Public Health (FOPH–BAG) under the license number A202819 and was performed in the biosafety level 3 laboratory at the Institute of Virology and Immunology (IVI), Mittelhäusern, Switzerland, under appropriate safety measures. Work with the replication-competent chimeric virus VSV*ΔG-S$_{\Delta21}$ has been approved by the FOPH under the license number A130982 and was performed at the IVI under biosafety level 2 conditions.

## Data availability

The plasmids encoding for the six highest affinity binders are available through Addgene (Addgene #153522, #153523, #153524, #153525, #153526, and #153527). Purified Sb-Fc constructs can be commercially obtained from Absolute Antibody. The three-dimensional cryo-EM density maps have been deposited in the Electron Microscopy Data Bank under accession numbers EMD-12082 (http://www.ebi.ac.uk/pdbe/entry/EMD-12082) (SARS-CoV-2 spike protein in complex with Sb#15 and Sb#68 in a 3up conformation), EMD-12083 (http://www.ebi.ac.uk/pdbe/entry/EMD-12083) (SARS-CoV-2 spike protein in complex with Sb#15 and Sb#68 in a 1up/1up-out/1down conformation), EMD-12084 (http://www.ebi.ac.uk/pdbe/entry/EMD-12084) (SARS-CoV-2 spike protein in complex with Sb#15 in a 1up/1up-out/1down conformation), EMD-12085 (http://www.ebi.ac.uk/pdbe/entry/EMD-12085) (SARS-CoV-2 spike protein in complex with Sb#68 in a 2up/1flexible conformation), and EMD-12086 (http://www.ebi.ac.uk/pdbe/entry/EMD-12086) (SARS-CoV-2 spike protein in complex with Sb#68 in a 1up/2down conformation) and include the cryo-EM maps, both half-maps, the unmasked and unsharpened refined maps, and the mask used for final FSC calculation. Coordinates of the models have been deposited in the Protein Data Bank. The accession numbers are 7P77 (http://www.rcsb.org/pdb/explore/explore.do?structureId=7P77), 7P78 (http://www.rcsb.org/pdb/explore/explore.do?structureId=7P78), 7P79 (http://www.rcsb.org/pdb/explore/explore.do?structureId=7P79), 7P7A (http://www.rcsb.org/pdb/explore/explore.do?structureId=7P7A), and 7P7B (http://www.rcsb.org/pdb/explore/explore.do?structureId=7P7B), respectively. The NGS data of the escape mutation experiment were deposited in the ENA databank under accession ID: PRJEB49553 (http://www.ebi.ac.uk/ena/data/view/PRJEB49553).

**Expanded View** for this article is available online.

### Acknowledgements

We thank Rony Nehmé and André Heuer (Creoptix AG, Wädeswil, Switzerland) for the acquisition, fitting and interpretation of a first set of GCI measurements using the WAVEsystem. We thank Florence Projer, David Hacker and Kelvin Lau (Protein Production and Structure Core Facility, EPFL, Switzerland) for the production of the prefusion spike protein. We are grateful to Jason McLellan (The University of Texas at Austin, U.S.) for having provided the pre-fusion-stabilized soluble spike expression vectors for S-2P and S-6P. We thank Michael Fiebig (Absolute Antibody) for providing us with purified Sb-Fc. We thank Raimund Dutzler and Marta Sawicka (University of Zurich) for freezing cryo-EM grids. Michiel Punter (University of Groningen) is acknowledged for IT help. We are also grateful to the Next Generation Sequencing Platform, University of Bern for their contribution to this study. This work was funded by the SNSF in the frame of the National Research Programme Covid-19 (NRP 78) (Project number 4078P0_198314). Open access funding provided by Universitat Zurich.

## Author contributions

**Justin Walter:** Conceptualization; Supervision; Investigation; Visualization; Methodology; Writing—original draft; Writing—review & editing. **Melanie Scherer:** Validation; Investigation; Visualization; Methodology; Writing—review & editing. **Cedric A J Hutter:** Software; Investigation; Visualization; Methodology; Writing—review & editing. **Alisa A Garaeva:** Software; Validation; Investigation; Visualization; Methodology; Writing—review & editing. **Iwan Zimmermann:** Investigation. **Marianne Wyss:** Investigation. **Jan Rheinberger:** Resources; Software; Investigation. **Yelena Ruedin:** Investigation. **Jennifer C Earp:** Investigation. **Pascal Egloff:** Investigation. **Michèle Sorgenfrei:** Investigation. **Lea Hürlimann:** Investigation. **Imre Gonda:** Investigation. **Gianmarco Meier:** Investigation. **Sille Remm:** Investigation. **Sujani Thavarasah:** Investigation. **Geert van Geest:** Resources; Software. **Rémy Bruggmann:** Resources; Software. **Gert Zimmer:** Supervision; Investigation; Methodology; Writing—review & editing. **Dirk Jan Slotboom:** Supervision; Writing—review & editing. **Cristina Paulino:** Supervision; Funding acquisition; Investigation; Writing—original draft; Writing—review & editing. **Philippe Plattet:** Conceptualization; Funding acquisition; Investigation; Visualization; Writing—original draft; Project administration; Writing—review & editing. **Markus Seeger:** Conceptualization; Supervision; Funding acquisition; Investigation; Visualization; Writing—original draft; Project administration; Writing—review & editing.

In addition to the CRediT author contributions listed above, the contributions in detail are:
JDW, PP, and MAS conceived the project. JDW and IZ cloned and purified target proteins. CAJH performed the sybody selection, made the bispecific constructs, and purified sybodies. JDW and CAJH performed ELISA and GCI experiments. AAG and JR acquired and processed cryo-EM data under the supervision of CP and DJS using proteins prepared by JDW and CAJH. MS and MW performed neutralization assays under the supervision of PP using pseudotyped VSV and sybodies produced by CAJH and JDW. YR performed the neutralization experiments with live SARS-CoV-2 under the supervision of GZ. GZ performed escape mutation experiments, which were analyzed by NGS by GvG and RB. JCE prepared GFP-purification resin. PE, MS, and IZ were involved in planning the sybody selection strategy. LMH, IG, GM, SR, and ST discussed and interpreted data. JDW, CAJH, AAG, MS, and MAS prepared figures. JDW, CP, PP, and MAS wrote the paper. MS, CAJH, AAG, GZ, and DJS made major contributions in the course of paper editing.

## Disclosure and competing interests statement

IZ, PE, and MAS are co-founders and shareholders of Linkster Therapeutics AG. The other authors do not declare any conflict of interest.

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
