## [Review Process File · EMBO Reports]

Biparatopic antibodies neutralize SARS-CoV-2 variants of concern and mitigate drug resistance

Justin Walter, Melanie Scherer, Cedric Hutter, Alisa Garaeva, Iwan Zimmermann, Marianne Wyss, Jan Rheinberger, Yelena Ruedin, Jennifer Earp, Pascal Egloff, Michèle Sorgenfrei, Lea Hürliemann, Imre Gonda, Gianmarco Meier, Silke Remm, Sujani Thavarasah, Geert van Geest, Rémy Bruggmann, Gert Zimmer, Dirk Slotboom, Cristina Paulino, Philippe Plattet, and Markus Seeger

DOI: 10.15252/embr.202154199

Corresponding author(s): Markus Seeger (m.seeger@imm.uzh.ch), Philippe Plattet (philippe.plattet@vetsuisse.unibe.ch), Cristina Paulino (c.paulino@rug.nl)

Review Timeline:

Transfer from Review	21st Oct 21
Commons: Editorial Decision:	22nd Oct 21
Revision Received:	11th Jan 22
Editorial Decision:	19th Jan 22
Revision Received:	27th Jan 22
Accepted:	31st Jan 22

Editor: Achim Breiling

Transaction Report: This manuscript was transferred to EMBO Reports following peer review at Review Commons

**Review
COMMONS**

Dear Dr. Seeger,

Thank you for transferring your manuscript from Review Commons to EMBO reports. I now went through your manuscript, the referee reports (attached again below), and your revision plan. All referees acknowledge the high interest of the findings. They have raised a number of suggestions to improve the manuscript, which I see you are able to address during a minor revision of the manuscript.

We would thus like to invite you to revise your manuscript with the understanding that all referee points must be addressed in the revised manuscript or in the detailed point-by-point response (as suggested in your revision plan). Acceptance of your manuscript will depend on a positive outcome of a second round of review. It is EMBO reports policy to allow a single round of revision only and acceptance of the manuscript will therefore depend on the completeness of your responses included in the next, final version of the manuscript.

Revised manuscripts should be submitted within three months of a request for revision. Please contact me to discuss the revision should you need additional time.

1) a .docx formatted version of the final manuscript text (including legends for main figures, EV figures and tables), but without the figures included. Please make sure that changes are highlighted to be clearly visible. Figure legends should be compiled at the end of the manuscript text.

2) individual production quality figure files as .eps, .tif, .jpg (one file per figure), of main figures and EV figures. Please upload these as separate, individual files upon re-submission.

The Expanded View format, which will be displayed in the main HTML of the paper in a collapsible format, has replaced the Supplementary information. You can submit up to 5 images as Expanded View. Please follow the nomenclature Figure EV1, Figure EV2 etc. The figure legend for these should be included in the main manuscript document file in a section called Expanded View Figure Legends after the main Figure Legends section. Additional Supplementary material should be supplied as a single pdf file labeled Appendix. The Appendix should have page numbers and needs to include a table of content on the first page (with page numbers) and legends for all content. Please follow the nomenclature Appendix Figure Sx, Appendix Table Sx etc. throughout the text, and also label the figures and tables according to this nomenclature. Movies and Datasets should be uploaded separately, using the nomenclature Movie EVx and Dataset EVx. Please provide a legend with title for each movie file as text file and upload it ZIPed together with the movie file. For datasets, please provide a title and legend on the first TAB of the excel file.

See also our guide for figure preparation:

4) a complete author checklist, which you can download from our author guidelines

(<https://www.embopress.org/page/journal/14693178/authorguide>). Please insert page numbers in the checklist to indicate where the requested information can be found in the manuscript. The completed author checklist will also be part of the RPF.

5) that primary datasets produced in this study (e.g. RNA-seq, ChIP-seq, structural and array data) are deposited in an appropriate public database. If no primary datasets have been deposited, please also state this a dedicated section (e.g. 'No primary datasets have been generated and deposited'), see below.

The accession numbers and database should be listed in a formal "Data Availability " section (placed after Materials & Methods) that follows the model below. This is now mandatory (like the COI statement). Please note that the Data Availability Section is restricted to new primary data that are part of this study.

Data availability

6) We strongly encourage the publication of original source data with the aim of making primary data more accessible and transparent to the reader. The source data will be published in a separate source data file online along with the accepted manuscript and will be linked to the relevant figure. If you would like to use this opportunity, please submit the source data (for example scans of entire gels or blots, data points of graphs in an excel sheet, additional images, etc.) of your key experiments together with the revised manuscript. If you want to provide source data, please include size markers for scans of entire gels, label the scans with figure and panel number, and send one PDF file per figure.

8) Regarding data quantification and statistics, can you please specify, where applicable, the number "n" for how many independent experiments (biological replicates or technical replicates - please clearly indicate this) were performed, the bars and error bars (e.g. SEM, SD) and the test used to calculate p-values in the respective figure legends. Please provide statistical testing where applicable, and also add a paragraph detailing this to the methods section. See: <http://www.embopress.org/page/journal/14693178/authorguide#statisticalanalysis>

9) Please note our reference format:

10) Please add up to 5 key words to the title page.

11) Please provide the abstract written in present tense.

12) Please add a conflict-of-interest statement to the manuscript text and order the manuscript sections like this:
Title page - Abstract - Introduction - Results - Discussion - Materials and Methods - DAS - Acknowledgements - Author contributions - Conflict of interest - References - Figure legends - Expanded View Figure legends

In addition, I would need from you:

- a short, two-sentence summary of the manuscript (not more than 35 words).
- two to four short bullet points highlighting the key findings of your study.
- a schematic summary figure (in jpeg or tiff format with the exact width of 550 pixels and a height of not more than 400 pixels) that can be used as a visual synopsis on our website.

Finally, please note that all corresponding authors are required to supply an ORCID ID for their name upon submission of a revised manuscript. Please ask your co-corresponding author Philippe Plattet to do that. Please find instructions on how to link

the ORCID ID to the account in our manuscript tracking system in our author guidelines:
<http://www.embopress.org/page/journal/14693178/authorguide#authorshipguidelines>

I look forward to seeing a revised version of your manuscript when it is ready. Please let me know if you have questions or comments regarding the revision.

Please use this link to submit your revision: <https://embor.msubmit.net/cgi-bin/main.plex>

Yours sincerely,

Achim Breiling
Editor
EMBO Reports

Referee #1:

Two Sybodies have been selected against RBD. These bind to non-overlapping epitopes. Cryo EM and X-Ray crystallography was used to find the recognition epitopes on the trimeric antigen and to reveal conformational variance. Each antibody modestly neutralising the virus or viral particles, however, the biparatopic construct were highly neutralising infectivity. Also escape variants were no longer generated when viruses were challenged with the biparatopic construct.

The paper is written in a clear and logical manner.

**** Significance ****

Data are convincing, all experimental details are disclosed. The conclusions are based on solid data.
Significance (Required) This paper gives us deep-insight in possible strategies to avoid the emergence of SARS-Cov2 mutants, while treating victims of Covid19.

****Referees cross-commenting****

All three reviewers agree that this is solid work and deserves publication after minor amendments (rephrasing) and without the need of additional experiments. All remarks raised can be easily answered (within one or two weeks time). The extra explanations that are requested are probably already available in the drawers of the authors (or can be provided after quick consultation of literature to have the digits right).

Referee #2:

****Summary****

The manuscript by Walter et al. describes the identification and characterisation of synthetic nanobodies (Sybodies) reactive against the SARS-CoV-2 RBD. Two leading candidates binding non-overlapping epitopes were selected and engineered into first a biparatopic construct, then a "trimer of dimers", with binding affinity for the RBD being increased by each engineering stage. Though for SB#15, neutralisation potency against VSV pseudotyped with the SARS-CoV-2 spike protein was reduced for the Beta variant, the biparatopic construct GS4 and particularly the Trimer of dimers, showed high neutralising potency against VSV pseudotyped with the SARS-CoV-2 Alpha and Beta variant spikes, overcoming the sensitivity to variants of the individual nanobodies.

****Major Comments****

1.Line 187: The increase in potency seen here for the Biparatopic GS4 construct is described as due to a highly synergistic mechanism, the increase seen with co-administration of SB#15 and SB#68 as individual entities seemed to be additive rather than displaying synergy. How do the authors distinguish increased potency due to avidity effects in GS4 and a "synergistic" mechanism? Could they please describe this or remove the statement.

2.Line 336-338: The authors report here that neutralisation values of the sybodies described is comparable to those isolated from immune libraries using phage display. Nanobodies have been described in several papers from these platforms that neutralise much more potently than those described here, for example 0.1 nM in a PRNT assay (Pymm et al. 2021), 0.022 nM in PRNT and 0.045 nM in pseudovirus assays (Xiang et al. 2020) as compared with >2000 nM for both SB#15 and SB#68 in PRNT and >140 nM in pseudovirus assays.

Of the references used here, Koenig et al also describe neutralisation values in their PRNT assay that range from ~50-150 nM rather than the >2000 nM seen here, and 2-3 fold more potent neutralisation values in their pseudovirus assay. Wagner describes neutralisation values of 7 nM and Hanke report pseudovirus neutralisation at ~50 nM (again approximately 2-3 fold more potent).

Could the authors please discuss the neutralisation potency in the context of these values as they do not seem comparable to many of the nanobodies generated from immune libraries. There are also other examples of sybodies in neutralisation assays for SARS-CoV-2, do the neutralisation values measured in these studies add to the argument that these platforms are comparable?

****Minor Comments****

1.Line 233: The tripod design shown here appears to contain trimerization domains from parainfluenza virus and T4 bacteriophage. Do the authors anticipate problems with immune reactivity to these domains in therapeutic settings and how could this be mitigated if so? Could the authors please add a line to the discussion addressing this?

2.Line 52-55: Many papers have started to detail the antibody binding landscape for RBD reactive antibodies and nanobodies in SARS-CoV-2 (Barnes, West et al., 2020; Dejnirattisai et al., 2021, Sun, D et al. 2021; Wheatley, AK et al. 2021 and others). Are the percentage of antibodies targeting these two "hotspots" known in-vivo and if so, could this be included to better reflect the diversity of antibodies characterised to date?

3.Line 106-108: Could the authors please include whether they thought the improvement seen here was an additive or synergistic effect if known?

4.Line 171: Could the authors please add a figure call out for S7B here.

5.Line 174: Could the authors please add a figure call out for S7D and E here.

6.Line 176: Figure S7B is called out here, but the text appears to be referring to figure S7C. Please could the authors correct this and alter figure S7 so that the figures are called out in order (as S7D and E have already been described in the text).

7.Line 293: WNb 10 (Pymm et al.) which also binds in this region was additionally shown to cross recognise SARS-CoV-1. Please could the authors mention the cross-recognition of the WNb 10 nanobody with SARS-CoV-1 here.

8.Line 609: For Cryo-EM studies, the complex is described as using a 1:1.3 molar ratio of Spike to sybody, given that each spike contains three RBD moieties that are potential binding sites for the sybody, is the ratio given here correct, or does it refer to the ratio for each RBD?

9.Figure 3: Could the authors please add labels for both D and E to show regions of the spike monomer, e.g. RBD, NTD, S1, S2 etc. and the angular displacement of the up-out RBD conformation from the up-RBD.

10.Figure 5A legend: The colour code described for figure 5A should be reversed (the salmon spheres seem to refer to the global variants, not the adaptation experiments).

11.Figures S5-S7: The RBD labels in these figures are in each case directly over the NTD. Could an arrow be added or positioning altered to improve clarity?

**** Significance ****

This is a considerable body of high-quality work, though of course is in an extremely rapidly moving field. Nanobodies capable of neutralising SARS-CoV-2 virus in vitro, both alone and as cocktail combinations have been widely described with detailed structural work to define their epitopes and interaction with residues mutated in the variants. Indeed, the affinities of SB#15 and SB#68 for WT SARS-CoV-2 RBD and the mutations contained within the Alpha and Beta variants, as well as the crystal structure for SB#68 have previously been published (Ahmad et al. 2021). The authors acknowledge this through comparison of their nanobodies and the epitopes they target with others in the field, many of which have also demonstrated efficacy in preventing SARS-CoV-2 infection in in vivo models.

The manuscript defines a novel orientation of the RBD within the spike trimer presumably driven by the binding of these nanobodies to both the RBD involved and adjacent RBD's, though it is not clear if this unique orientation has any direct impact on the stability of the spike trimer and mechanism of neutralisation.

The combination of the two sybodies into a biparatopic format, and the incorporation of these biparatopic moieties into a trimeric construct is a novel aspect of the paper, as previous trimers have involved either single nanobodies joined into a trimer, or single nanobodies displayed on a similar "tripod" structure (Güttler et al. 2021). This format demonstrates an ability to compensate for

the sensitivity of component nanobodies to RBD variation and to considerably increase neutralising potency, though the ability of this construct to neutralise SARS-CoV-2 in vivo and to be translated into a viable therapeutic remains to be addressed.

Reviewer field of expertise: Structural biology, infection and immunity.

Referee #3:

****Summary:****

The manuscript 'Biparatopic sybody constructs neutralize SARS-CoV-2 variants of concern and mitigate emergence of drug resistance' by Walter et al. describes and characterizes a pair of synergistic SARS-CoV-2 neutralizing sybodies. Sybodies are proteins derived from synthetic libraries designed based on variable domains of camelid heavy chain-only antibodies. The authors characterize binding properties of the two sybodies Sb#15 and Sb#68 and quantify their neutralization potential against SARS-CoV-2 spike-pseudotyped VSV and authentic virus. They revealed synergy between both sybodies and systematically followed up the improvement of neutralizing activity by biparatopic fusions (Sb#15-linker-Sb#68), Fc fusions leading to homodimerization (Sb#15-Fc; Sb#68-Fc), as well as an entirely new hexameric configuration realized by a trimerization domain fused to bi-paratopic Sb#68-linker-Sb#15. The latter arrangement ultimately improved neutralization by more than 1000-fold and thus may serve a general blueprint for the improvement of antiviral sybodies or nanobodies.

The authors also determined the structural basis of neutralization and found that only the combination of both sybodies stabilized the RBD 3-up conformation of spike, which may block Ace2 binding and induce the pre-mature rearrangement (and thus inactivation) of spike as postulated previously. A novel configuration with one RBD up, one RBD up-and-out, and one RBD down (a sofar not described configuration) was found in the presence of both sybodies as well as Sb#15 alone.

The authors further confirm by in vitro evolution experiments that the combination of sybodies targeting two epitopes prevents the emergence of escape variants, while experiments with single nanobodies revealed possible escape variants that had in part already been identified in patients. In line with the multiple epitopes targeted, the bi- and multi-valent sybodies retained the capacity to neutralize emerging SARS-CoV-2 variants.

****Major comments****

The data of this manuscript is of high quality. The claims are supported by solid data and no further experiments are required to back up the claims. The data is presented in a transparent matter and replicates and statistical analyses are appropriate for all experiments.

****Minor suggestions for data presentation:****

As neutralization experiments and the resulting IC50 values differ from system to system and lab to lab, it would be helpful to include on additional reagent for neutralization that is described in other publications and would help to compare values. This could be (commercially available) ACE2-Fc or any other nanobody or sybody with published neutralization potential. While all the IC50 values of neutralization are described in table 3, it may be helpful to also show them in the respective graphs themselves.

**** Significance ****

This is the latest in a long series of antibodies, nanobodies, and sybodies neutralizing SARS-CoV-2 by binding to the RBD of spike ((Güttler et al., 2021; Hanke et al., 2020; Huo et al., 2020; Koenig et al., 2021; Lv et al., 2020; Schoof et al., 2020; Wrapp et al., 2020; Xiang et al., 2020). In fact, one of the first (and fastest) publications on SARS-CoV-2 specific sybodies is from the authors of this manuscript themselves, although this first description of SARS sybodies is only published on BioRxiv (Walter et al., 2020). Now the authors picked up one pair of particularly interesting synergistic sybodies and analyzed them in detail. Few other publications have characterized nanobodies in that degree of functional, evolutionary, structural, and mechanistic detail. It turns out that the epitopes of both nanobodies as well as the likely mechanism of action is shared with a similar pair of nanobodies derived from immunized camelids, and likely also with more combinations of nanobodies binding to these dominant epitopes (Koenig et al., 2021). While the study does not provide novel mechanistic insight per se, in particular the structural information will be helpful to deduce common mechanisms of synergistic neutralization. Importantly, the authors have developed an entirely novel format to trimerize bi-paratopic nanobodies, which improves neutralization even more and has the potential to be applied for neutralizing nanobody against this and other viruses (or other receptors).

To reveal the structural basis of synergistic neutralization, and likely gain insights into coronavirus fusion itself, it will require a number of structures as the one described here. In particular the cryo EM structures of both nanobodies bound to the ectodomain of spike as well as the confirmed stabilization of the RBD 3-up configuration will thus be of value for the field. Few studies determine sybody- or nanobody-specific escape variants as described here, and the data will therefore also be of value for more systematic assessments of antigenic escape.

The study itself is interesting for a broader audience interested in virology, antibody responses (and evolution), nanobody technology and translational aspect of sybodies and other biologics derived from antibodies.

My expertise is based on long standing research in virology and nanobody development. While I can interpret the structures of nanobody-target complexes on a functional level, my expertise is not sufficient to judge the technical aspects of the solution of electron microscopy structures themselves.

- Güttler, T., Aksu, M., Dickmanns, A., Stegmann, K. M., Gregor, K., Rees, R., Taxer, W., Rymarenko, O., Schünemann, J., Dienemann, C., Gunkel, P., Mussil, B., Krull, J., Teichmann, U., Groß, U., Cordes, V. C., Döbelstein, M., & Görlich, D. (2021). Neutralization of SARS-CoV-2 by highly potent, hyperthermostable, and mutation-tolerant nanobodies. *The EMBO Journal*, e107985. <https://doi.org/10.15252/EMBJ.2021107985>
- Hanke, L., Vidakovics Perez, L., Sheward, D. J., Das, H., Schulte, T., Moliner-Morro, A., Corcoran, M., Achour, A., Karlsson Hedestam, G. B., Hällberg, B. M., Murrell, B., & McInerney, G. M. (2020). An alpaca nanobody neutralizes SARS-CoV-2 by blocking receptor interaction. *Nature Communications*, 11(1), 4420. <https://doi.org/10.1038/s41467-020-18174-5>
- Huo, J., le Bas, A., Ruza, R. R., Duyvesteyn, H. M. E., Mikolajek, H., Malinauskas, T., Tan, T. K., Rijal, P., Dumoux, M., Ward, P. N., Ren, J., Zhou, D., Harrison, P. J., Weckener, M., Clare, D. K., Vogirala, V. K., Radecke, J., Moynié, L., Zhao, Y., ... Naismith, J. H. (2020). Neutralizing nanobodies bind SARS-CoV-2 spike RBD and block interaction with ACE2. *Nature Structural & Molecular Biology*, 1-9. <https://doi.org/10.1038/s41594-020-0469-6>
- Koenig, P.-A., Das, H., Liu, H., Kümmerer, B. M., Gohr, F. N., Jenster, L.-M., Schiffelers, L. D. J., Tesfamariam, Y. M., Uchima, M., Wuerth, J. D., Gatterdam, K., Ruetalo, N., Christensen, M. H., Fandrey, C. I., Normann, S., Tödtmann, J. M. P., Pritzl, S., Hanke, L., Boos, J., ... Schmidt, F. I. (2021). Structure-guided multivalent nanobodies block SARS-CoV-2 infection and suppress mutational escape. *Science (New York, N.Y.)*. <https://doi.org/10.1126/science.abe6230>
- Lv, Z., Deng, Y.-Q. Q., Ye, Q., Cao, L., Sun, C.-Y. Y., Fan, C., Huang, W., Sun, S., Sun, Y., Zhu, L., Chen, Q., Wang, N., Nie, J., Cui, Z., Zhu, D., Shaw, N., Li, X.-F. F., Li, Q., Xie, L., ... Wang, X. (2020). Structural basis for neutralization of SARS-CoV-2 and SARS-CoV by a potent therapeutic antibody. *Science*, 369(6509), 1505-1509. <https://doi.org/10.1126/SCIENCE.ABC5881>
- Schoof, M., Faust, B., Saunders, R. A., Sangwan, S., Rezelj, V., Hoppe, N., Boone, M., Billesbølle, C. B., Zimanyi, M., Deshpande, I., Liang, J., Anand, A. A., Dobzinski, N., Zha, B. S., Barsi-Rhyne, B., Belyy, V., Barile-Hill, A. W., Gupta, S., Simoneau, C. R., ... Manglik, A. (2020). An ultra-high affinity synthetic nanobody blocks SARS-CoV-2 infection by locking Spike into an inactive conformation. *BioRxiv*, 10(5), 2020.08.08.238469. <https://doi.org/10.1101/2020.08.08.238469>
- Walter, J. D., Hutter, C. A. J., Zimmermann, I., Earp, J., Egloff, P., Sorgenfrei, M., Hürlimann, L. M., Gonda, I., Meier, G., Remm, S., Thavarasah, S., Plattet, P., & Seeger, M. A. (2020). Synthetic nanobodies targeting the SARS-CoV-2 receptor-binding domain. *BioRxiv*, 2020.04.16.045419. <https://doi.org/10.1101/2020.04.16.045419>
- Wrapp, D., de Vlieger, D., Corbett, K. S., Torres, G. M., Wang, N., van Breedam, W., Roose, K., van Schie, L., Hoffmann, M., Pöhlmann, S., Graham, B. S., Callewaert, N., Schepens, B., Saelens, X., & McLellan, J. S. (2020). Structural basis for potent neutralization of betacoronaviruses by single-domain camelid antibodies. *Cell*, 181(5), 1004-1015.e15. <https://doi.org/10.1016/j.cell.2020.04.031>
- Xiang, Y., Nambulli, S., Xiao, Z., Liu, H., Sang, Z., Duprex, W. P., Schneidman-Duhovny, D., Zhang, C., & Shi, Y. (2020). Versatile and multivalent nanobodies efficiently neutralize SARS-CoV-2. *Science*, 370(6523), eabe4747. <https://doi.org/10.1126/science.abe4747>

****Referees cross-commenting****

I concur with reviewer 1. This solid piece of work is ready for publication after minor text changes and does not require further experiments. The authors should discuss their claim of synergy and comment on the potency of neutralization (ideally with some data on a molecule whose potency was also quantified by other systems).

Point-by-point answer to the reviewer comments

We thank the reviewers for their constructive feedback and we have revised the manuscript accordingly.

Reviewer #1 (Evidence, reproducibility and clarity (Required)):

Two Sybodies have been selected against RBD. These bind to non-overlapping epitopes. Cryo EM and X-Ray crystallography was used to find the recognition epitopes on the trimeric antigen and to reveal conformational variance. Each antibody was modestly neutralising the virus or viral particles, however, the biparatopic construct were highly neutralising infectivity. Also escape variants were no longer generated when viruses were challenged with the biparatopic construct.

The paper is written in a clear and logical manner.

Data are convincing, all experimental details are disclosed. The conclusions are based on solid data.

Reviewer #1 (Significance (Required)):

This paper gives us deep-insight in possible strategies to avoid the emergence of SARS-Cov2 mutants, while treating victims of Covid19.

****Referees cross-commenting****

All three reviewers agree that this is solid work and deserves publication after minor amendments (rephrasing) and without the need of additional experiments.

All remarks raised can be easily answered (within one or two weeks time). The extra explanations that are requested are probably already available in the drawers of the authors (or can be provided after quick consultation of literature to have the digits right).

Reviewer 1 (Serge Muyldermans).

We thank Reviewer 1 for this positive assessment of our manuscript.

Reviewer #2 (Evidence, reproducibility and clarity (Required)):

****Summary****

The manuscript by Walter et al. describes the identification and characterisation of synthetic nanobodies (Sybodies) reactive against the SARS-CoV-2 RBD. Two leading candidates binding non-overlapping epitopes were selected and engineered into first a biparatopic construct, then a "trimer of dimers", with binding affinity for the RBD being increased by each engineering stage. Though for SB#15, neutralisation potency against VSV pseudotyped with the SARS-CoV-2 spike protein was reduced for the Beta variant, the biparatopic construct GS4 and particularly the Trimer of dimers, showed high neutralising potency against VSV pseudotyped with the SARS-CoV-2 Alpha and Beta variant spikes, overcoming the sensitivity to variants of the individual nanobodies.

****Major Comments****

1.Line 187: The increase in potency seen here for the Biparatopic GS4 construct is described as due to a highly synergistic mechanism, the increase seen with co-administration of SB#15 and SB#68 as individual entities seemed to be additive rather than displaying synergy. How do the authors distinguish increased potency due to avidity effects in GS4 and a "synergistic" mechanism? Could they please describe this or remove the statement.

We agree with the reviewer's concern. Indeed, our data do not support synergy of Sb#15 and Sb#68, because (as the reviewer rightly points out) the increase of potency is minimal when these sybodies are added as individual entities. Rather, our data support a mechanism in which the increased potency of GS4 is solely attributed to the avidity effect of the construct. We have thus removed this statement in line 187 and throughout the rest of the manuscript.

2.Line 336-338: The authors report here that neutralisation values of the sybodies described is comparable to those isolated from immune libraries using phage display. Nanobodies have been described in several papers from these platforms that neutralise much more potently than those described here, for example 0.1 nM in a PRNT assay (Pymm et al. 2021), 0.022 nM in PRNT and 0.045 nM in pseudovirus assays (Xiang et al. 2020) as compared with >2000 nM for both SB#15 and SB#68 in PRNT and >140 nM in pseudovirus assays.

We believe that reviewer#2 misunderstood/misinterpreted our statement.

In line 336-338, we state:

"In contrast to a number of synthetic or naïve SARS-CoV-2 nanobodies from other libraries that required a post-selection maturation process to reach satisfactory affinities [28, 40-42], sybodies selected by us and by other labs [36, 43] exhibited affinities in the single and double digit nM range and where thus of similar affinity as nanobodies isolated from immune libraries using classical phage display [11, 37, 44]."

Hence, our statement relates to binder affinities (and not neutralization values, which appear to differ quite a lot among different studies). In addition, we explicitly mention that the affinities of the sybodies (single digit nM/low double digit nM) are in a similar range as nanobodies isolated from immune library using classical phage display (i.e. with a similar standard-screening approach via ELISA), as can be read/looked up in the cited papers [11, 37, 44].

Although it is correct that the affinities of the nanobodies described in Pymm et al. 2021 and Xiang et al. 2020 were found to be stronger than sybodies (i.e. a good number of picomolar binders identified), one has to take into consideration that the immunization/screening regime was quite extensive for Pymm et al. and that in Xiang et al., a massive deep-screen using NGS and mass spectrometry was performed to identify highly affine nanobodies.

If selection and screening efforts on par with Pymm et al. and Xiang et al. had been performed with sybodies, we are convinced that picomolar binders would have been identified as well.

For these reasons, we consider our original statement as adequate.

Of the references used here, Koenig et al also describe neutralisation values in their PRNT assay that range from ~50-150 nM rather than the >2000 nM seen here, and 2-3 fold more potent neutralisation values in their pseudovirus assay. Wagner describes neutralisation values of 7 nM and Hanke report pseudovirus neutralisation at ~50 nM (again approximately 2-3 fold more potent).

Could the authors please discuss the neutralisation potency in the context of these values as they do not seem comparable to many of the nanobodies generated from immune libraries. There are also other examples of sybodies in neutralisation assays for SARS-CoV-2, do the neutralisation values measured in these studies add to the argument that these platforms are comparable?

We agree with reviewer#2 that the gap between PRNT and VSV neutralization assays were rather large in our case. Therefore, we repeated all our PRNT assays and part of the VSV neutralization assays. Further, we attempted to include the well-characterized antibody EY6A as well as the sybody MR3 (<https://www.nature.com/articles/s41467-021-24905-z>) as control.

In the repeated experiments of the PRNT assay, the potency of our sybodies and sybody constructs are indeed better, and the respective neutralization values between PRNT and VSV are now less apart (around 3-6 fold).

To explain this discrepancy, we added the following sentence to the text (line 118):

The approximately 3 to 6-fold discrepancy in neutralization efficacies, measured using either live SARS-CoV-2 virus or pseudotyped VSV, may reflect slight differences in viral physiology (variation of incorporated spikes per viral particle) or could owe to the different assay methods (luciferase emission versus plaque reduction determination).

Further, we were able to determine the neutralization values for the control sybody MR3, which was found to be identical to a value published in a previous study for the VSV assay.

Unfortunately, a commercially acquired EY6A antibody was found to be inactive in our PRNT assay for unclear reasons.

Nevertheless, we feel that we have sufficiently validated our neutralization assays, because their main purpose was not to exactly determine absolute neutralization values, but rather relative values among the different sybodies and their fusion constructs.

****Minor Comments****

1.Line 233: The tripod design shown here appears to contain trimerization domains from parainfluenza virus and T4 bacteriophage. Do the authors anticipate problems with immune reactivity to these domains in therapeutic settings and how could this be mitigated if so? Could the authors please add a line to the discussion addressing this?

The rationale behind the Tripod design was to investigate whether multivalency could further improve the potency of the biparatopic sybody construct. Our data indeed revealed that Tripod reached high neutralization efficiency against all tested SARS-CoV-2 variant of concerns (picomolar

range). We wish to note that even against pseudoviruses harboring spikes with mutations that disturbed single sybody's efficacy (e.g. Sb#15), IC₅₀s for the Tripod construct were still highly potent, thereby spotlighting the power of combining multivalency with the biparatopic strategy.

To facilitate trimerization of the biparatopic construct, we fused the T4 bacteriophage-derived foldon sequence as well as a mutated peptide derived from the trimeric canine distemper virus fusion (CDV-F) protein, a strategy that proved to be very effective.

We nevertheless fully agree with this reviewer's comment that such domains may affect drug's efficacy *in vivo* by, for instance, triggering undesired anti-drug immunity.

We added the following cautionary statement to the discussion (line 344):

“While our own Tripod-GS4r construct may cause problems with immune reactivity in a therapeutic setting due to the viral origin of the utilized trimerization domain, trimerization domains of human origin can be used instead to overcome this potential issue (Guttler et al, 2021).”

For *in vivo* experiments and potential future clinical treatments in humans, it would indeed be wise to exchange the CDV-F peptide and the foldon motif by a well-characterized trimerization domain derived from a human protein. As recently demonstrated (<https://doi.org/10.15252/emj.2021107985>), the NC1 trimerization motif of collagen XVIII is an excellent candidate (10.1016/j.jmb.2009.07.057).

We have added a sentence to the discussion addressing this important point.

As pointed out by reviewer #3, such a strategy may act as a general blueprint for the improvement of antiviral sybodies/nanobodies in the future.

2.Line 52-55: Many papers have started to detail the antibody binding landscape for RBD reactive antibodies and nanobodies in SARS-CoV-2 (Barnes, West et al., 2020; Dejnirattisai et al., 2021, Sun, D et al. 2021; Wheatley, AK et al. 2021 and others). Are the percentage of antibodies targeting these two "hotspots" known in-vivo and if so, could this be included to better reflect the diversity of antibodies characterised to date?

To date, most characterized antibodies were selected based on their ability to bind the RBD/NTD and neutralize SARS-CoV-2 infection. This is reflected in studies which aim to systematically categorize panels of antibodies/nanobodies against SARS-CoV-2 (KM Hastie, Science (2021):

<https://www.science.org/doi/10.1126/science.abh2315>; D Sun, et al, Bioarxiv (2021):

<https://www.biorxiv.org/content/10.1101/2021.03.09.434592v1.full>;

Y Wu, et al, Cell Host & Microb. (2020): [https://www.cell.com/cell-host-microbe/fulltext/S1931-](https://www.cell.com/cell-host-microbe/fulltext/S1931-3128(20)30250-X)

3128(20)30250-X;L Liu et al, Nature (2020): <https://www.nature.com/articles/s41586-020-2571-7>).

Due to this selection bias, it is difficult to assess the true overall “natural” percentage of antibodies targeting the two RBD “hotspots” in vivo. Available surveys of patient immune responses (E Shrock, et al, Science (2020):

<https://www.science.org/doi/10.1126/science.abd4250>; AK Wheatley, Cell

Reports (2021): <https://www.sciencedirect.com/science/article/pii/S2211124721012869>; AS Heffron,

et al, Plos Biol (2021):

<https://journals.plos.org/plosbiology/article?id=10.1371/journal.pbio.3001265>) tend to focus on a qualitative description of the entire epitope landscape, giving a perspective of overall epitope diversity yet still not enabling reliable quantification of percentages of antibodies targeting particular hotspots. With these points in mind, a recent report characterizing 179 anti-RBD monoclonal antibodies found that 83% (148/179) directly competed with ACE2 for RBD binding, whereas 17% (31/179) either did not compete with ACE2 or were found to be below the assay threshold to qualify as ACE2 blockers (KM Hastie 2021).

In summary, we feel that the question of “hotspots” is a very complicated one, which cannot be addressed with our own data. Therefore, we feel that it does not make sense to comment on this issue/topic in the discussion of our paper.

3.Line 106-108: Could the authors please include whether they thought the improvement seen here was an additive or synergistic effect if known?

As pointed out above, we have removed our statement regarding synergy, because our data does not support a synergistic action of the two sybodies.

4.Line 171: Could the authors please add a figure call out for S7B here.

We have re-organized Figure S7, as suggested.

5.Line 174: Could the authors please add a figure call out for S7D and E here.

We have re-organized Figure S7, as suggested.

6.Line 176: Figure S7B is called out here, but the text appears to be referring to figure S7C. Please could the authors correct this and alter figure S7 so that the figures are called out in order (as S7D and E have already been described in the text).

We have re-organized Figure S7, as suggested.

7.Line 293: WNb 10 (Pymm et al.) which also binds in this region was additionally shown to cross recognise SARS-CoV-1. Please could the authors mention the cross-recognition of the WNb 10 nanobody with SARS-CoV-1 here.

The mentioned paper (Pymm et al.) does not contain data supporting the cross-reactivity of WNb 10 with SARS-CoV-1. But its epitope (akin to Sb#68) overlaps with the one of VHH72 (originally selected against SARS-CoV-1) and therefore, WNb 10 likely cross-reacts.

Just one sentence down, we take reference to Pymm et al. Further, we show WNb 10 in the context of Figure 7. Hence, we feel that this nanobody is appropriately mentioned and referenced.

8.Line 609: For Cryo-EM studies, the complex is described as using a 1:1.3 molar ratio of Spike to sybody, given that each spike contains three RBD moieties that are potential binding sites for the sybody, is the ratio given here correct, or does it refer to the ratio for each RBD?

We thank the reviewer for having spotted this. The 1.3-fold molar excess of sybody was added relative to the spike monomer concentration, so this refers to the ratio for each RBD. We have changed the text accordingly.

9. Figure 3: Could the authors please add labels for both D and E to show regions of the spike monomer, e.g. RBD, NTD, S1, S2 etc. and the angular displacement of the up-out RBD conformation from the up-RBD.

This was changed accordingly.

10. Figure 5A legend: The colour code described for figure 5A should be reversed (the salmon spheres seem to refer to the global variants, not the adaptation experiments).

Thanks a lot for spotting this error, which we in the meantime corrected.

11. Figures S5-S7: The RBD labels in these figures are in each case directly over the NTD. Could an arrow be added or positioning altered to improve clarity?

This was changed accordingly.

Reviewer #2 (Significance (Required)):

This is a considerable body of high-quality work, though of course is in an extremely rapidly moving field. Nanobodies capable of neutralising SARS-CoV-2 virus *in vitro*, both alone and as cocktail combinations have been widely described with detailed structural work to define their epitopes and interaction with residues mutated in the variants. Indeed, the affinities of SB#15 and SB#68 for WT SARS-CoV-2 RBD and the mutations contained within the Alpha and Beta variants, as well as the crystal structure for SB#68 have previously been published (Ahmad et al. 2021).

It is worth mentioning that we were among the first groups making nanobody sequences freely available to the science community. The team around Ahmad et al. gained access to the sybodies via Addgene and independently characterized them at the structural level. Importantly, these confirmatory analyses (published recently in JBC) in fact re-inforce our findings and add an additional layer of validation. Finally, it should be added that we have determined here cryo-EM structures (not performed in Ahmad et al) and we made the bi-paratopic and tripod constructs and characterized them in terms of neutralization potency and viral escape mutations.

The authors acknowledge this through comparison of their nanobodies and the epitopes they target with others in the field, many of which have also demonstrated efficacy in preventing SARS-CoV-2 infection in *in vivo* models.

The manuscript defines a novel orientation of the RBD within the spike trimer presumably driven by the binding of these nanobodies to both the RBD involved and adjacent RBD's, though it is not clear if this unique orientation has any direct impact on the stability of the spike trimer and mechanism of neutralisation.

The combination of the two sybodies into a biparatopic format, and the incorporation of these biparatopic moieties into a trimeric construct is a novel aspect of the paper, as previous trimers have

involved either single nanobodies joined into a trimer, or single nanobodies displayed on a similar "tripod" structure (Güttler et al. 2021). This format demonstrates an ability to compensate for the sensitivity of component nanobodies to RBD variation and to considerably increase neutralising potency, though the ability of this construct to neutralise SARS-CoV-2 in vivo and to be translated into a viable therapeutic remains to be addressed.

Reviewer field of expertise: Structural biology, infection and immunity.

Reviewer #3 (Evidence, reproducibility and clarity (Required)):

****Summary:****

The manuscript 'Biparatopic sybody constructs neutralize SARS-CoV-2 variants of concern and mitigate emergence of drug resistance' by Walter et al. describes and characterizes a pair of synergistic SARS-CoV-2 neutralizing sybodies. Sybodies are proteins derived from synthetic libraries designed based on variable domains of camelid heavy chain-only antibodies. The authors characterize binding properties of the two sybodies Sb#15 and Sb#68 and quantify their neutralization potential against SARS-CoV-2 spike-pseudotyped VSV and authentic virus. They revealed synergy between both sybodies and systematically followed up the improvement of neutralizing activity by biparatopic fusions (Sb#15-linker-Sb#68), Fc fusions leading to homodimerization (Sb#15-Fc; Sb#68-Fc), as well as an entirely new hexameric configuration realized by a trimerization domain fused to bi-paratopic Sb#68-linker-Sb#15. The latter arrangement ultimately improved neutralization by more than 1000-fold and thus may serve a general blueprint for the improvement of antiviral sybodies or nanobodies.

The authors also determined the structural basis of neutralization and found that only the combination of both sybodies stabilized the RBD 3-up conformation of spike, which may block Ace2 binding and induce the pre-mature rearrangement (and thus inactivation) of spike as postulated previously. A novel configuration with one RBD up, one RBD up-and-out, and one RBD down (a sofar not described configuration) was found in the presence of both sybodies as well as Sb#15 alone.

The authors further confirm by in vitro evolution experiments that the combination of sybodies targeting two epitopes prevents the emergence of escape variants, while experiments with single nanobodies revealed possible escape variants that had in part already been identified in patients. In line with the multiple epitopes targeted, the bi- and multi-valent sybodies retained the capacity to neutralize emerging SARS-CoV-2 variants.

****Major comments****

The data of this manuscript is of high quality. The claims are supported by solid data and no further experiments are required to back up the claims. The data is presented in a transparent matter and replicates and statistical analyses are appropriate for all experiments.

We thank reviewer#3 for this very positive assessment.

****Minor suggestions for data presentation:****

As neutralization experiments and the resulting IC50 values differ from system to system and lab to lab, it would be helpful to include an additional reagent for neutralization that is described in other publications and would help to compare values. This could be (commercially available) ACE2-Fc or any other nanobody or sybody with published neutralization potential.

As pointed out above in our response to reviewer #2, we addressed this concern with the following experiments:

We repeated all our PRNT assays and part of the VSV neutralization assays. Further, we attempted to include the well-characterized antibody EY6A as well as the sybody MR3 (<https://www.nature.com/articles/s41467-021-24905-z>) as control.

In the repeated experiments of the PRNT assay, the potency of our sybodies and sybody constructs are indeed better, and the respective neutralization values between PRNT and VSV are now less apart (around 3-6 fold).

To explain this discrepancy, we added the following sentence to the text (line 118):

The approximately 3 to 6-fold discrepancy in neutralization efficacies, measured using either live SARS-CoV-2 virus or pseudotyped VSV, may reflect slight differences in viral physiology (variation of incorporated spikes per viral particle) or could owe to the different assay methods (luciferase emission versus plaque reduction determination).

Further, we were able to determine the neutralization values for the control sybody MR3, which was found to be identical to a value published in a previous study for the VSV assay.

Unfortunately, a commercially acquired EY6A antibody was found to be inactive in our PRNT assay for unclear reasons.

Nevertheless, we feel that we have sufficiently validated our neutralization assays, because their main purpose was not to exactly determine absolute neutralization values, but rather relative values among the different sybodies and their fusion constructs.

While all the IC50 values of neutralization are described in table 3, it may be helpful to also show them in the respective graphs themselves.

We tried to include these values directly in the graphs, but this looked confusing and was difficult to read. Therefore, we kept it as it was.

Reviewer #3 (Significance (Required)):

This is the latest in a long series of antibodies, nanobodies, and sybodies neutralizing SARS-CoV-2 by binding to the RBD of spike ((Güttler et al., 2021; Hanke et al., 2020; Huo et al., 2020; Koenig et al., 2021; Lv et al., 2020; Schoof et al., 2020; Wrapp et al., 2020; Xiang et al., 2020). In fact, one of the first (and fastest) publications on SARS-CoV-2 specific sybodies is from the authors of this manuscript themselves, although this first description of SARS sybodies is only published on BioRxiv (Walter et

al., 2020). Now the authors picked up one pair of particularly interesting synergistic sybodies and analyzed them in detail. Few other publications have characterized nanobodies in that degree of functional, evolutionary, structural, and mechanistic detail. It turns out that the epitopes of both nanobodies as well as the likely mechanism of action is shared with a similar pair of nanobodies derived from immunized camelids, and likely also with more combinations of nanobodies binding to these dominant epitopes (Koenig et al., 2021). While the study does not provide novel mechanistic insight per se, in particular the structural information will be helpful to deduce common mechanisms of synergistic neutralization. Importantly, the authors have developed an entirely novel format to trimerize bi-paratopic nanobodies, which improves neutralization even more and has the potential to be applied for neutralizing nanobody against this and other viruses (or other receptors).

To reveal the structural basis of synergistic neutralization, and likely gain insights into coronavirus fusion itself, it will require a number of structures as the one described here. In particular the cryo EM structures of both nanobodies bound to the ectodomain of spike as well as the confirmed stabilization of the RBD 3-up configuration will thus be of value for the field. Few studies determine sybody- or nanobody-specific escape variants as described here, and the data will therefore also be of value for more systematic assessments of antigenic escape.

The study itself is interesting for a broader audience interested in virology, antibody responses (and evolution), nanobody technology and translational aspect of sybodies and other biologics derived from antibodies.

My expertise is based on long standing research in virology and nanobody development. While I can interpret the structures of nanobody-target complexes on a functional level, my expertise is not sufficient to judge the technical aspects of the solution of electron microscopy structures themselves.

Güttler, T., Aksu, M., Dickmanns, A., Stegmann, K. M., Gregor, K., Rees, R., Taxer, W., Rymarenko, O., Schünemann, J., Dienemann, C., Gunkel, P., Mussil, B., Krull, J., Teichmann, U., Groß, U., Cordes, V. C., Dobbelsstein, M., & Görlich, D. (2021). Neutralization of SARS-CoV-2 by highly potent, hyperthermostable, and mutation-tolerant nanobodies. *The EMBO Journal*, e107985.

<https://doi.org/10.15252/EMBJ.2021107985>

Hanke, L., Vidakovic Perez, L., Sheward, D. J., Das, H., Schulte, T., Moliner-Morro, A., Corcoran, M., Achour, A., Karlsson Hedestam, G. B., Hällberg, B. M., Murrell, B., & McInerney, G. M. (2020). An alpaca nanobody neutralizes SARS-CoV-2 by blocking receptor interaction. *Nature Communications*, 11(1), 4420. <https://doi.org/10.1038/s41467-020-18174-5>

Huo, J., le Bas, A., Ruza, R. R., Duyvesteyn, H. M. E., Mikolajek, H., Malinauskas, T., Tan, T. K., Rijal, P., Dumoux, M., Ward, P. N., Ren, J., Zhou, D., Harrison, P. J., Weckener, M., Clare, D. K., Vogirala, V. K., Radecke, J., Moynié, L., Zhao, Y., ... Naismith, J. H. (2020). Neutralizing nanobodies bind SARS-CoV-2 spike RBD and block interaction with ACE2. *Nature Structural & Molecular Biology*, 1-9.

<https://doi.org/10.1038/s41594-020-0469-6>

Koenig, P.-A., Das, H., Liu, H., Kümmerer, B. M., Gohr, F. N., Jenster, L.-M., Schiffelers, L. D. J., Tesfamariam, Y. M., Uchima, M., Wuerth, J. D., Gatterdam, K., Ruetalo, N., Christensen, M. H., Fandrey, C. I., Normann, S., Tödtmann, J. M. P., Pritzl, S., Hanke, L., Boos, J., ... Schmidt, F. I. (2021). Structure-guided multivalent nanobodies block SARS-CoV-2 infection and suppress mutational escape. *Science (New York, N.Y.)*. <https://doi.org/10.1126/science.abe6230>

Lv, Z., Deng, Y.-Q. Q., Ye, Q., Cao, L., Sun, C.-Y. Y., Fan, C., Huang, W., Sun, S., Sun, Y., Zhu, L., Chen, Q.,

Wang, N., Nie, J., Cui, Z., Zhu, D., Shaw, N., Li, X.-F. F., Li, Q., Xie, L., ... Wang, X. (2020). Structural basis for neutralization of SARS-CoV-2 and SARS-CoV by a potent therapeutic antibody. *Science*, 369(6509), 1505-1509. <https://doi.org/10.1126/SCIENCE.ABC5881>

Schoof, M., Faust, B., Saunders, R. A., Sangwan, S., Rezelj, V., Hoppe, N., Boone, M., Billesbølle, C. B., Zimanyi, M., Deshpande, I., Liang, J., Anand, A. A., Dobzinski, N., Zha, B. S., Barsi-Rhyne, B., Belyy, V., Barile-Hill, A. W., Gupta, S., Simoneau, C. R., ... Manglik, A. (2020). An ultra-high affinity synthetic nanobody blocks SARS-CoV-2 infection by locking Spike into an inactive conformation. *BioRxiv*, 10(5), 2020.08.08.238469. <https://doi.org/10.1101/2020.08.08.238469>

Walter, J. D., Hutter, C. A. J., Zimmermann, I., Earp, J., Egloff, P., Sorgenfrei, M., Hürlimann, L. M., Gonda, I., Meier, G., Remm, S., Thavarasah, S., Plattet, P., & Seeger, M. A. (2020). Synthetic nanobodies targeting the SARS-CoV-2 receptor-binding domain. *BioRxiv*, 2020.04.16.045419. <https://doi.org/10.1101/2020.04.16.045419>

Wrapp, D., de Vlieger, D., Corbett, K. S., Torres, G. M., Wang, N., van Breedam, W., Roose, K., van Schie, L., Hoffmann, M., Pöhlmann, S., Graham, B. S., Callewaert, N., Schepens, B., Saelens, X., & McLellan, J. S. (2020). Structural basis for potent neutralization of betacoronaviruses by single-domain camelid antibodies. *Cell*, 181(5), 1004-1015.e15. <https://doi.org/10.1016/j.cell.2020.04.031>

Xiang, Y., Nambulli, S., Xiao, Z., Liu, H., Sang, Z., Duprex, W. P., Schneidman-Duhovny, D., Zhang, C., & Shi, Y. (2020). Versatile and multivalent nanobodies efficiently neutralize SARS-CoV-2. *Science*, 370(6523), eabe4747. <https://doi.org/10.1126/science.abe4747>

****Referees cross-commenting****

I concur with reviewer 1. This solid piece of work is ready for publication after minor text changes and does not require further experiments. The authors should discuss their claim of synergy and comment on the potency of neutralization (ideally with some data on a molecule whose potency was also quantified by other systems).

Dear Dr. Seeger

Thank you for the submission of your revised manuscript to our editorial offices. I now went through your detailed p-b-p-response letter, and I consider the concerns by the referees as adequately addressed.

Before we can proceed with formal acceptance, I have these editorial requests I ask you to address in a final revised manuscript:

- Please shorten the title to not more than 100 characters (including spaces).
- Per journal policy, we do not allow 'data not shown', which is stated in the manuscript (page 14). All data referred to in the paper should be displayed in the main or Expanded View figures, or an Appendix. Thus, please add these data (or change the text accordingly if these data are not central to the study). See:
<http://embor.embopress.org/authorguide#unpublisheddata>
- Please name the 'Data availability statement' 'Data availability section'.
- Please order the manuscript sections like this (using these names):
Title page - Abstract - Introduction - Results - Discussion - Materials and Methods - Data availability section - Acknowledgements - Author contributions - Conflict of interest statement - References - Figure legends - Expanded View Figure legends - Tables with legends
- Please add links to directly access the datasets mentioned in the 'Data availability section'.
- The data deposited at the ENA databank under ID PRJEB49553 cannot be found there. Please make sure the dataset is deposited, that the accession ID is correct, and that the data is public (latest upon publication of the manuscript).
- Please make sure that all figure panels and tables (also those shown in the Appendix) are called out and that they are called out sequentially. Presently, Fig 4D is called out after 6B, and there seem to be no callouts for Fig. EV1B, the separate panels of Fig. EV3 and for panels A, B and D of Fig. EV5. Also callouts for Appendix items are missing. Please check and in case change the order of the panels in the figures.
- Please add a full table of contents including page numbers to the Appendix file. It is also not necessary to repeat here the full title page of the paper.
- It seems that Lea Hürlimann, Imre Gonda, Gianmarco Meier, Sille Remm, Sujani Thavarasah, Geert van Geest and Remy Bruggmann are missing from the author contributions. Please check.
- Thanks for providing the synopsis image. Could you provide this with fonts in bold/filled? Presently, the writing looks a bit fuzzy and weak.

Finally, We updated our journal's competing interests policy in January 2022 and request authors to consider both actual and perceived competing interests. Please review the policy (see: <https://www.embopress.org/competing-interests>) and update your competing interests if necessary.

In addition, I would need from you:

- a short, two-sentence summary of the manuscript (not more than 35 words).
- two to four short (2 lines) bullet points highlighting the key findings of your study.

Best,

The authors have addressed all minor editorial requests.

Markus Seeger
Institute of Medical Microbiology, University of Zurich
Switzerland

Dear Dr. Seeger,

I am very pleased to accept your manuscript for publication in the next available issue of EMBO reports. Thank you for your contribution to our journal.

At the end of this email I include important information about how to proceed. Please ensure that you take the time to read the information and complete and return the necessary forms to allow us to publish your manuscript as quickly as possible.

As part of the EMBO publication's Transparent Editorial Process, EMBO reports publishes online a Review Process File to accompany accepted manuscripts. As you are aware, this File will be published in conjunction with your paper and will include the referee reports, your point-by-point response and all pertinent correspondence relating to the manuscript.

If you do NOT want this File to be published, please inform the editorial office within 2 days, if you have not done so already, otherwise the File will be published by default [contact: emboreports@embo.org]. If you do opt out, the Review Process File link will point to the following statement: "No Review Process File is available with this article, as the authors have chosen not to make the review process public in this case."

Thank you again for your contribution to EMBO reports and congratulations on a successful publication. Please consider us again in the future for your most exciting work.

Yours sincerely,

Achim Breiling
Editor
EMBO Reports

THINGS TO DO NOW:

You will receive proofs by e-mail approximately 2-3 weeks after all relevant files have been sent to our Production Office; you should return your corrections within 2 days of receiving the proofs.

Please inform us if there is likely to be any difficulty in reaching you at the above address at that time. Failure to meet our deadlines may result in a delay of publication, or publication without your corrections.

All further communications concerning your paper should quote reference number EMBOR-2021-54199V3 and be addressed to emboreports@wiley.com.

Should you be planning a Press Release on your article, please get in contact with emboreports@wiley.com as early as possible, in order to coordinate publication and release dates.

Corresponding Author Name: Markus A. Seeger

Manuscript Number: EMBOR-2021-54199V1